# More of Less: A Rashomon Algorithm for Sparse Model Sets

## Abstract

The current paradigm of machine learning consists in finding a single best model to deliver predictions and, if possible, interpretations for a specific problem. This paradigm has however been strongly challenged in recent years through the study of the *Rashomon Effect* which was coined initially by Leo Breiman. This phenomenon occurs when there exist many good predictive models for a given dataset/problem, with considerable practical implications in terms of interpretation, usability, variable importance, replicability and many others. The set of models (within a specific class of functions) which respect this definition is referred to as the *Rashomon set* and an important amount of recent work has been focused on ways of finding these sets as well as studying their properties. Developed in parallel to current research on the Rashomon Effect and motivated by sparse latent representations for high-dimensional problems, we present a heuristic procedure that aims to find sets of sparse models with good predictive power through a greedy forward-search that explores the low-dimensional variable space. Throughout this algorithm, good low-dimensional models identified from the previous steps are used to build models with more variables in the following steps. While not directly targeting Rashomon sets, empirical findings show that these sparse model sets preserve almost-equal performance with respect to a single reference model in a given class (i.e. a Rashomon set) and include diverse models which can be combined into networks that deliver additional layers of interpretation and new insights into how variable combinations can explain the Rashomon Effect.

## 1 Introduction

The purpose of any machine learning algorithm is to deliver precise predictions with respect to a response (or responses) of interest given a set of variables (predictors), whether dealing with classification or more general regression problems. The current paradigm to achieve such predictions is commonly based on a *single* best model that has been chosen, parametrized and fine-tuned to address a specific problem or dataset. There are many reasons why such a paradigm is the dominating one, among which (i) its reliance on the existence of a unique optimal representation of the data generation process as well as (ii) the practical implications of relying on a single set of parameters and variables that can be used, and eventually interpreted, for predictions on similar problems. There are however various limitations with this paradigm, starting from areas of research and practice where there is a need for flexibility in the variables used. Indeed, from genomics (see e.g. Xiong et al., 2001) to online prediction (see e.g. Carmona-Cejudo et al., 2011), there are many tasks where a multitude of subsets of variables are useful, such as (i) in medical studies where machines collect different measurements (variables) for a specific problem (see e.g. Draghici et al., 2006); (ii) for online search algorithms where every subject provides different variables (according to their preferences or willingness to disclose information) to determine suggestions or matches (see e.g. Vaughan & Chen, 2015) ; (iii) in signal processing and pattern recognition where signals and images are collected at different resolutions and therefore a single representation may not be flexible enough to adapt to different signal and image features (see e.g. Elad & Yavneh, 2009; Wang et al., 2018). Intuitively, there can indeed be circumstances (e.g. medical diagnostics) where some variables are more costly to measure and practitioners would therefore prefer to have alternative good models with more "accessible" variables. In addition, interpretability of phenomena can be greatly enhanced and stabilized when considering a set of models as highlighted by the multimodel

inference[1] and model selection uncertainty literature in areas such as sociology and, especially, ecology (see e.g. Burnham & Anderson, 2004; Anderson & Burnham, 2004; Harrison et al., 2018; Caruana et al., 2004, for an overview). This is of particular relevance, for example, within the Predictability-Computability-Stability (PCS) framework of Yu & Kumbier (2020).

The existence of multiple models that fit a specific problem/dataset similarly well was already posited in Breiman (2001b), albeit for more fundamental reasons coming from the uncertainty and noise affecting data. Indeed, as a result of this uncertainty, many approximately-equally-good models can exist for a specific problem/dataset giving rise to the so-called *Rashomon Effect* (Breiman, 2001b). While being commonly observed in practice, this phenomenon returned to be an area of fundamental research only in recent years starting from the work of Fisher et al. (2019) where they develop a new metric of variable importance using multiple good models, and followed by a growing focus on this topic in Semenova et al. (2022); Xin et al. (2022); Zhong et al. (2024); Liu et al. (2022); Qinyu Zhu et al. (2023); Semenova et al. (2024); Kissel & Mentch (2024) where they develop methods and measures to select and evaluate sets of good models belonging to different model classes. The need for a paradigm shift in machine learning as a result of this effect was underlined more recently in Rudin et al. (2024) where the authors also highlighted the numerous advantages that can be obtained when finding sets of models, as opposed to a single one, especially for high-stakes decision making. In particular these advantages include, but are not limited to: (i) the possibility for users to determine their preferred representation of a problem without losing predictive power; (ii) stable variable importance information; and (iii) novel insights for model interpretability and prediction uncertainty. As a consequence, there is a need to develop procedures to find these sets of models which, intuitively, are called *Rashomon sets* (Fisher et al., 2019; Semenova et al., 2022). The push to develop such procedures is very recent and, currently, exist for decision trees (Xin et al., 2022), generalized additive models (Zhong et al., 2024) and risk scores (Liu et al., 2022; Qinyu Zhu et al., 2023). More specifically, having defined a class of models, these procedures are able to rapidly find the corresponding (approximate) Rashomon sets which achieve a predictive performance within an $\theta$-range of a reference model.

In this work we continue the current effort of studying the Rashomon Effect by delivering a specific procedure that aims at selecting a set (library) of good predictive models under the *constraint of sparsity for high-dimensional problems* and for any class of functions defined by the user. While this procedure does not aim to find Rashomon sets (and therefore does not respect their definition), it is nevertheless related to them through the search of sets of good performing *sparse* models. More specifically, the motivation behind this new procedure lies in another possible explanation for the existence of the Rashomon Effect for high-dimensional settings, namely the presence of *latent (unobserved) variables* that generate, or are associated to, the manifest variables used in practice to predict the response (which is actually linked to the latent variables). Indeed, the existence of the Rashomon Effect has mainly been explained this far by the presence of nondeterministic processes in the data which are affected by noise and uncertainty (Semenova et al., 2024). This indeed can explain how different model parametrizations from the same class, or even how models from different classes, can be considered almost-equally good. However, the presence of latent variables can also explain how models (in a given class) can contain different combinations of manifest variables that all have similar predictive performance based on how well these combinations represent the underlying latent (true) model. This is in line with Fisher et al. (2019) where they state that, even if the models in the set do not contain the true data generating process (e.g. the latent structure), one can hope that some models in the set can work in similar ways to it. While the study of latent variable modelling is vast (see e.g. Borsboom, 2008; Muthén & Muthén, 2009), including sparse latent representations (see e.g. Wu et al., 2022; Ahuja et al., 2022; Fumero et al., 2023), the idea of studying sparse latent structures through sets of good models is novel and can provide new ways of representing these problems. In particular, since "*the Rashomon Effect gives rise to simpler-yet-accurate models*" (Rudin et al., 2024), we focus on finding sparse representations of high-dimensional problems by finding sets of sparse models containing combinations of fewer variables which also have the advantage of delivering more interpretable models (Rudin, 2019; Rudin et al., 2022). In addition, the proposed procedure is class-independent since it consists in a wrapper heuristic in which the user can select their class of preference (similarly to Guerrier et al., 2016; Kissel & Mentch, 2024, for example). More specifically, the proposed procedure performs variable screening procedures and greedily combines the screened variables to deliver

---

[1]Not to be confused with multimodal inference.

larger sets of sparse models. We call this procedure the "Sparse Wrapper AlGorithm" (SWAG) since any model class can be run within it (hence a wrapper) and it outputs a set of sparse representations for these models with good predictive performance. In particular, as highlighted earlier, this method delivers one class of possible perturbations recommended within the PCS framework put forward in Yu & Kumbier (2020) and therefore contributes to model-stability (see also Kissel & Mentch, 2024). The following section describes the proposed algorithm, followed by various applied examples highlighting its advantages for different model classes and datasets.

## 2   A Sparse Rashomon Algorithm

Let us start with a general overview of the SWAG. This algorithm proceeds in a forward-stepwise manner: it builds and tests models starting from one variable until it includes a maximal number of variables per model, increasing the number of variables at each step. Hence, for each fixed number of variables, the algorithm tests various (randomly selected) models and picks those with the best performance in terms of a chosen performance metric (e.g. test error estimates). Throughout, the algorithm uses the information coming from the best models at the previous step to build and test models in the following step. In the end, it outputs a set of sparse models with similar performance to the best one in the set. Given its nature, a more detailed comparison of this algorithm with the current stepwise selection procedures can be found in App. A.

To now provide a more formal and detailed description, let us define some basic notation. Let $\mathbf{y} \in \mathbb{R}^n$ denote the response and $\mathbf{X} \in \mathbb{R}^{n \times p}$ denote a variable (design) matrix with $n$ samples and $p$ variables, the latter being indexed by a set $\mathcal{S} := \{1, \ldots, p\}$. In addition, we denote a class of functions (i.e. models) as $\mathcal{L} := \mathcal{L}(\mathbf{y}, \mathbf{X})$ with $l \in \mathcal{L}$ denoting a general model which is built by using a subset $s_l \in \mathcal{P}(\mathcal{S})$ of variables in $\mathbf{X}$, where we let $\mathcal{P}(\mathcal{A})$ and $|\mathcal{A}|$ denote respectively the power set and cardinality of a set $\mathcal{A}$. In the following paragraphs we will proceed to describing the algorithm and introduce meta-parameters whose interpretation and selection will be discussed later in Sec. 2.1. Also, based on this description, Sec. 2.2 will compare the SWAG library of models to Rashomon sets. It must also be noted that, given the heuristic nature of the algorithm, many of its steps described below can be adapted/modified according to the user's needs. This being said, the first choice to make for the SWAG is to determine the maximum dimension of variables that the user wants to be considered in a model and we denote this parameter as $p_{\max} < p$. Based on this parameter, the SWAG aims at exploring the space of variables to find sets of models using $\hat{p}$ variables ($1 \le \hat{p} \le p_{\max}$) with low error (or good prediction). In this respect, without loss of generality, we assume the choice of a performance metric that should be minimized, i.e. an error $\epsilon$, such as estimates of test errors (e.g. cross-validation, Akaike Information Criterion etc.). With this in mind, the SWAG is described in the following paragraphs, where the first two algorithms are defined for a general model dimension $\hat{p}$ and are then used within the third algorithm which combines them in a greedy manner.

**First Step**   The first screening step starts by using one *distinct* variable at a time to create $p$ models. Once these models are built, a set of models $\mathcal{M}^\star$ is now available which is indexed by the ordered index set $\mathcal{I} := \{1, \ldots, p\}$ (i.e. each model $l \in \mathcal{M}^\star$ is indexed by a unique element $i \in \mathcal{I}$). Having chosen a performance error $\epsilon$, we denote the vector containing the errors of the $p$ one-dimensional variable models as $\boldsymbol{\epsilon}^\star \in \mathbb{R}^p$ which is also indexed by the set $\mathcal{I}$ (i.e. each model $l \in \mathcal{M}^\star$ is associated with an element in the error vector $\boldsymbol{\epsilon}^\star$). Given this, it is now possible to select a performance quantile $q_\alpha^\star$, where $\alpha \in (0,1)$ and the quantile estimator is chosen by the user (in this work we use the default choice for all statistical software, see e.g. Hyndman & Fan, 1996). The smaller the value of $\alpha$, the smaller the errors selected. The procedure then selects all the models whose error is smaller or equal to $q_\alpha^\star$ and includes these in a new model set $\widetilde{\mathcal{M}}^\star$. The set $\widetilde{\mathcal{M}}^\star$ therefore collects one-dimensional models (i.e. models with one variable) with small errors and are therefore based on a subset of variables $\mathcal{S}^\star \subset \mathcal{S}$ that can be assumed to be predictive with respect to the response of interest $\mathbf{y}$. Similarly to other screening procedures, we assume that this procedure is able to select predictive variables with high probability. This assumption however does not necessarily hold when there are predictive variables in the data that only demonstrate importance (i.e. predictive power) when combined in models with other variables: if they exist, these variables will not be evaluated in the next steps of the SWAG even though they would be important to better predict the response. To avoid this limitation, modifications can obviously be made, including a change of the screening criterion or a less strict exclusion of non-screened

variables in the following steps of the SWAG. Nevertheless, based on all examples presented in Sec. 3 (and in other applications), this limitation does not appear to affect the algorithm's performance compared to, for example, models that make use of all variables. This screening procedure is described in Algo. 1.

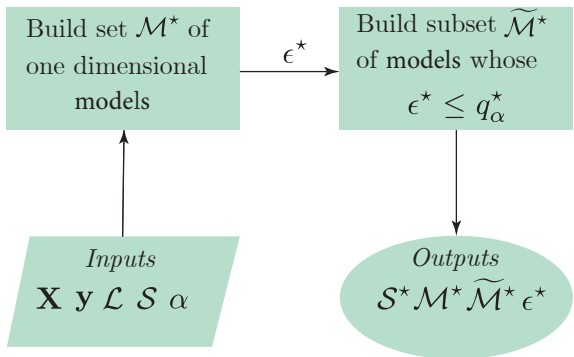

**Algorithm 1** First Screening Algorithm

INPUTS: A response $\mathbf{y} \in \mathbb{R}^n$ and variables $\mathbf{X} \in \mathbb{R}^{n \times p}$; A variable index set $\mathcal{S} := \{1, \dots, p\}$; A class of models $\mathcal{L}$; A performance percentile $\alpha \in (0, 1)$; Eventual parameters to compute the error $\epsilon$.

1: Using the class $\mathcal{L}$, build $p$ models by using all variables in the set $\mathcal{S}$
2: Create a model set $\mathcal{M}^\star$ (with $|\mathcal{M}^\star| = p$)
3: Build an error vector $\boldsymbol{\epsilon}^\star \in \mathbb{R}^p$ and identify the $\alpha$-quantile $q_\alpha^\star$ of this vector
4: Create new model set $\widetilde{\mathcal{M}}^\star$ with models whose error is smaller or equal to $q_\alpha^\star$
5: Create variable index set $\mathcal{S}^\star$ with variables included in the models in the set $\mathcal{M}^\star$

OUTPUTS: $\mathcal{S}^\star$; $\widetilde{\mathcal{M}}^\star$; $\mathcal{M}^\star$; $\boldsymbol{\epsilon}^\star$

**Remark.** *This first step is conceived for high-dimensional problems (i.e. large p). However, this step can be avoided entirely if the dimension of the data is reasonable. Indeed, we would ideally like to skip this first step and consider all variables (or at least force some additional variables) in the following steps of the SWAG. This would imply that $\widetilde{\mathcal{M}}^\star = \{l \in \mathcal{L}, \forall s_l \in \mathcal{S}\}$ and $\mathcal{S}^* = \mathcal{S}$ within the steps described below.*

**General Step** Fixing a given variable dimension $\hat{p}$ such that $2 \leq \hat{p} \leq p_{\max}$, the general screening step builds a maximum number $m$ of *distinct* models which will all be included in a model set $\mathcal{M}^{\hat{p}}$ where each model is built on combinations of $\hat{p}$ *distinct* variables. In order to build these $m$ models, the general step takes the variable index set $\mathcal{S}^\star$ from Algo. 1 as well as a set of models $\widehat{\mathcal{M}}$ where each model is of dimension $\hat{p} - 1$ (i.e. each model in $\widehat{\mathcal{M}}$ takes $\hat{p} - 1$ variables as an input). We let $s_l \in \tilde{S} := \{s \in \mathcal{P}(\mathcal{S}^\star) \,\big|\, |s| = \hat{p} - 1\}$ denote the variable indices for a specific model $l \in \widehat{\mathcal{M}}$. By randomly selecting a model $l \in \widehat{\mathcal{M}}$, the general step will build a new model by using the variables indexed by $s_l$ and a randomly selected distinct variable from the index set $\mathcal{S}^\star \setminus s_l$. This is repeated until $m$ distinct models are built and included in the model set $\mathcal{M}^{\hat{p}}$. It must be noted that the total number of distinct models that can be built this way, say $\tilde{m}$, can actually be smaller than $m$. In this case, the set $\mathcal{M}^{\hat{p}}$ will contain all possible models that can be built through this procedure. Once the candidate model set $\mathcal{M}^{\hat{p}}$ is built, the general step of the algorithm closely follows Algo. 1. More specifically, an error vector $\boldsymbol{\epsilon}^{\hat{p}}$ is built and its performance quantile $q_\alpha^{\hat{p}}$ is computed. The main output of this step is a new model set $\widetilde{\mathcal{M}}^{\hat{p}}$ which includes all models whose error in $\boldsymbol{\epsilon}^{\hat{p}}$ is smaller or equal to $q_\alpha^{\hat{p}}$ (with both $\boldsymbol{\epsilon}^{\hat{p}}$ and $\mathcal{M}^{\hat{p}}$ being ordered by the same index set $\mathcal{I}^{\hat{p}} := \{1, \dots, m\}$). This general step is described in Algo. 2.

The above described algorithms provide a straightforward manner of selecting a set of good models for a given variable dimension $\hat{p}$. However, as the variable dimension increases, the number of possible distinct variable combinations increases exponentially fast leading to an increased risk of inefficiently exploring the variable space if one simply randomly picks $m$ variable combinations. For this reason, the SWAG performs a *greedy* procedure which uses the information from Algo. 1 to obtain the set of best variables $\mathcal{S}^\star$ which is the easiest to explore completely. The next step of the algorithm, with $\hat{p} = 2$, then takes the set $\mathcal{S}^\star$ and the set of best models $\widetilde{\mathcal{M}}^\star$ as the input $\widehat{\mathcal{M}}$ for Algo. 2. At each of the following steps, with $\hat{p} > 2$, the algorithm defines $\widehat{\mathcal{M}} := \widetilde{\mathcal{M}}^{\hat{p}-1}$. Therefore, when increasing the variable dimension, the algorithm only considers variable combinations based on good models and variables from the previous dimension. This procedure is repeated for all variable dimensions until the maximal dimension $p_{\max}$ is reached. Throughout the procedure, the algorithm saves the good model sets and error vectors for each dimension $\hat{p}$. More formally, letting $l_i$ and $\boldsymbol{\epsilon}_i$ represent the $i^{th}$ model and its associated error metric respectively, the set of SWAG models can be therefore defined as:

$$\widetilde{\mathcal{M}} := \bigcup_{\hat{p}=1}^{p_{\max}} \left\{ l_i \in \mathcal{M}^{\hat{p}} \,\middle|\, \boldsymbol{\epsilon}_i^{\hat{p}} \leq q_\alpha^{\hat{p}}, \, \forall \, i \in \mathcal{I}^{\hat{p}} \right\}.$$

---

**Algorithm 2** General Screening Algorithm

---

INPUTS: A response $\mathbf{y} \in \mathbb{R}^n$ and variables $\mathbf{X} \in \mathbb{R}^{n \times p}$; A variable index set $\mathcal{S}^\star \subset \{1, \ldots, p\}$ from Algo. 1; A number of variables $\hat{p} \leq p_{\max}$; A model set $\widehat{\mathcal{M}}$; A class of models $\mathcal{L}$; A maximum number of models $m$; A performance percentile $\alpha \in (0, 1)$; Eventual parameters to compute the error $\epsilon$.

1: Define total possible $\hat{p}$-dimensonal models, i.e. $\tilde{m}$
2: **if** $\tilde{m} \leq m$ **then**
3:    Using the class $\mathcal{L}$, build all possible $\tilde{m}$ models with $\hat{p}$ variable inputs to create model set $\mathcal{M}^{\hat{p}}$
4: **else**
5:    Using the class $\mathcal{L}$, build $m$ models with $\hat{p}$ variable inputs by extracting $s_l$ from randomly sampled models $l \in \widehat{\mathcal{M}}$ and adding a variable from $\mathcal{S}^\star \setminus s_l$ to create model set $\mathcal{M}^{\hat{p}}$
6: **end if**
7: Build an error vector $\boldsymbol{\epsilon}^{\hat{p}}$ and identify the $\alpha$-quantile $q_\alpha^{\hat{p}}$ of this vector
8: Create new model set $\widetilde{\mathcal{M}}^{\hat{p}}$ with models whose error is smaller or equal to $q_\alpha^{\hat{p}}$

OUTPUTS: $\widetilde{\mathcal{M}}^{\hat{p}}$; $\mathcal{M}^{\hat{p}}$; $\boldsymbol{\epsilon}^{\hat{p}}$.

---

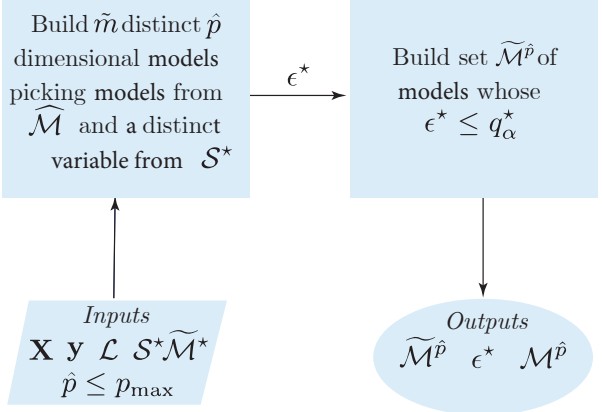

This procedure defines the SWAG which is described in Algo. 3. Let us now consider some aspects of interest for this algorithm, discussed in the following paragraphs.

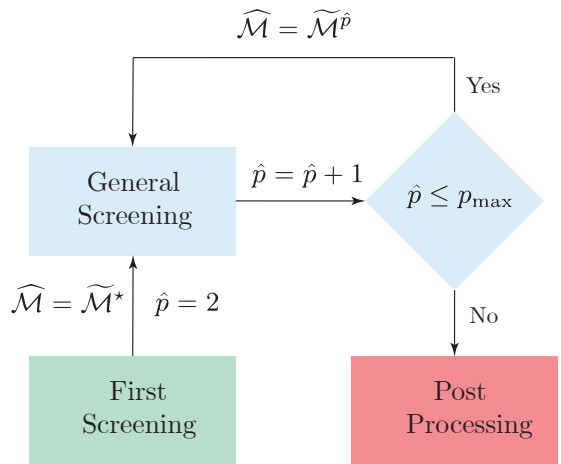

---

**Algorithm 3** SWAG

---

INPUTS: A response $\mathbf{y} \in \mathbb{R}^n$ and variables $\mathbf{X} \in \mathbb{R}^{n \times p}$; A variable index set $\mathcal{S} := \{1, \ldots, p\}$; A class of models $\mathcal{L}$; A maximum number of variables $p_{\max}$ $(< p)$; A maximum number of models $m$ for each step; A performance percentile $\alpha \in (0, 1)$; Eventual parameters to compute the error $\epsilon$.

1: Run Algo. 1 using inputs $\mathbf{y}$, $\mathbf{X}$, $\mathcal{S}$, $\mathcal{L}$, $m$, $\alpha$ and obtain $\mathcal{S}^\star$ and $\widetilde{\mathcal{M}}^\star$
2: $\widehat{\mathcal{M}} \leftarrow \widetilde{\mathcal{M}}^\star$
3: $\hat{p} \leftarrow 2$
4: **while** $\hat{p} \leq p_{\max}$ **do**
5:    Run Algo. 2 using inputs $\mathbf{y}$, $\mathbf{X}$, $\mathcal{S}^\star$, $\widehat{\mathcal{M}}$, $\mathcal{L}$, $m$, $\alpha$ and obtain $\widetilde{\mathcal{M}}^{\hat{p}}$
6:    $\widehat{\mathcal{M}} \leftarrow \mathcal{M}^{\hat{p}}$
7:    $\hat{p} \leftarrow \hat{p} + 1$
8: **end while**
9: Create
   • a set of good model sets $\widetilde{\mathcal{M}} := \{\widetilde{\mathcal{M}}^\star, \widetilde{\mathcal{M}}^2, \ldots, \widetilde{\mathcal{M}}^{p_{\max}}\}$
   • a set of error vectors $\tilde{\epsilon} := \{\epsilon^\star, \epsilon^2, \ldots, \epsilon^{p_{\max}}\}$

OUTPUTS: $\widetilde{\mathcal{M}}$; $\tilde{\epsilon}$

---

**Post-Processing** The user could choose to directly make use of the set of good models $\widetilde{\mathcal{M}}$ to make predictions based on different sets of variables or arrange variables into networks for interpretation and exploration (see for example Figs. 2 and 3 further on). However, these sets of models could undergo an additional screening procedure according to the needs of the user. For example, a possible approach that is used for the applications in Sec. 3 is to compute the median error for each vector in the set $\tilde{\epsilon}$ (as defined in Algo. 3) and select the quantile $\tilde{q}_\delta$ corresponding to the dimension whose median is the lowest, where $\delta \in (0, 1)$ can differ from $\alpha$ (a possible choice is 0.01). Having identified this quantile, we then select the models (with the desired dimensions) whose error is smaller or equal to this quantile. In particular, this procedure would deliver a set closer in spirit (but not equivalent) to the original definition of the Rashomon set which contains all models whose performance is within an $\theta$-range of the reference model (see Sec. 2.2 for the definition). For this specific choice of post-processing, the quantile $\tilde{q}_\delta$ would represent the upper bound

defining the performance of models in the Rashomon set, i.e.:

$$\widetilde{\mathcal{M}}_{\mathcal{R}} := \bigcup_{\hat{p}=1}^{p_{\max}} \left\{ l_i \in \mathcal{M}^{\hat{p}} \,\middle|\, \boldsymbol{\epsilon}_i^{\hat{p}} \leq \tilde{q}_\delta \,, \, \forall \, i \in \mathcal{I}^{\hat{p}} \right\}.$$

**Computational Complexity** The SWAG is a computationally intense procedure and builds at most $p + m\,(p_{\max} - 1)$ models in total which implies that its order of computation is proportional to this factor. In addition, the choice of the error and its computation can add complexity to the procedure (e.g. $r$-repeated $k$-fold cross-validation). Indeed, letting $C(n, \hat{p})$ represent the complexity of fitting a model $l \in \mathcal{L}$ to a dataset with $n$ samples and $\hat{p}$ variables, the worst-case complexity is given by $O\big( r\,k\,\big(p\,C(n,1) + p_{\max}\,m\,C(n, p_{\max})\big)\big)$. If a more efficient evaluation of the test error $\epsilon$ is available, this complexity can be considerably reduced. As an example, if using a generalized linear model with complexity $O(n\hat{p}^2 + \hat{p}^3)$ and using AIC or BIC as the error metric $\epsilon$, then the worst-case complexity becomes $O\left(np + mnp_{\max}^3 + mp_{\max}^4\right)$. One could even further reduce computational complexity by directly using training error metrics $\epsilon_{train}$ since the parameter $p_{\max}$ could already be considered as a regularizer to limit over-fitting (see Efron et al., 2004, and discussions in Sec. 2.1). A more detailed discussion of SWAG's computational complexity is found in App. B.

**Ensemble Learning and Stability** The goal of the SWAG does not consist in improving prediction accuracy *per se*, but more in providing user-flexibility and interpretability while preserving accuracy (to the best extent possible) by relying on individual models chosen from the diverse representations in the set $\widetilde{\mathcal{M}}$. Nevertheless, nothing stops us from using this set of good sparse models within an ensemble approach. For example the SWAG models could be included in Bagging (Breiman, 1996), Boosting (Schapire, 1990), Stacking (Wolpert, 1992) or other model-averaging approaches (see e.g., Raftery et al., 1997). In this sense, similarly to Kissel & Mentch (2024), the SWAG provides a procedure to define model perturbations as recommended, among others, within the PCS framework of Yu & Kumbier (2020).

**Limitations** Aside from the computational complexity and the previously mentioned limitation of the screening procedure for Algo. 1 (i.e. non-screened or excluded variables that would become important only when paired with other ones in the following dimensions), there are other clear limitations for the SWAG as is the case for many other heuristic approaches. More specifically, the SWAG attempts to approximate an exhaustive search up to dimension $p_{\max}$ by limiting the evaluated combinations to $m$ (for $\hat{p} \geq 2$). Therefore, there will always be a risk that an important combination of variables may not be evaluated at a given dimension. Due to the greedy forward nature of the SWAG, this risk is increased if some important variables are not screened or if certain combinations are not evaluated from the previous steps. However, assuming that the screening procedure retains all (or most) of the important variables, then there is a high chance that screened variables that are marginally less predictive can still be selected in higher dimensions since strong models from previous dimensions are increased with other screened variables at each step, allowing them to be paired with the variables that ensure that they express their joint importance.

## 2.1 SWAG Parameters

For the application of the SWAG, we assume the user has chosen a class of models $\mathcal{L}$ for which they would like to obtain sparse representations without losing significant predictive power. Based on this, the user has to define the meta-parameters of the algorithm which are (i) the maximum variable dimension $p_{\max}$; (ii) the maximum number of models $m$ to be built within each step and (iii) the performance percentile $\alpha$. Ideally, with unlimited computing power, the first two parameters would be as large as possible, i.e. $p_{\max} = p$ and $m = \binom{p}{\lceil \frac{p}{2} \rceil}$, leading to best subset selection. However, this defeats the purpose of the algorithm and therefore the decision of these parameters must be based mainly on interpretability/flexibility requirements as well as available computing power and time. Below are some rules-of-thumb for the choice of these parameters:

- $p_{\max}$: Fixing the available computing power and the efficiency in the computation of the model class $\mathcal{L}$, this parameter will depend on the total dimension of the problem $p$. Indeed, the goal of the SWAG is to find very sparse models and the parameter $p_{\max}$ can be viewed as a regularization parameter

(Efron et al., 2004). Therefore, even with very large $p$, one can always fix this parameter within a range of 5-20 (or smaller) for interpretability and/or replicability purposes. Another criterion, when working with binary classification problems, is to use the *Event Per Variable* (EPV) rule presented in Vittinghoff & McCulloch (2007) and discussed in van der Ploeg et al. (2014) (see Sec. 3 for example). This rule aims at limiting the number of variables included in models to ensure that enough data points are available in each dimension spanned by these variables, thereby preserving sufficient information to adequately estimate such models. In future work, this parameter can be implicitly determined by the algorithm based on the error quantile (or other metric) as a stopping-criterion thereby defining $p_{\max}$ as the variable dimension where the error curve stops decreasing significantly similarly to the scree plot in factor or principal component analysis (see e.g. Cattell, 1966).

- $m$: Fixing the available computing power and the efficiency in the computation of the model class $\mathcal{L}$, this parameter will determine the proportion of variable space that will be explored by the algorithm. We know that it depends on the size of the problem $p$ since we necessarily have $m \geq p$ for the screening step of Algo. 1. In addition, this parameter needs to be chosen considering the performance percentile $\alpha$: if $m$ is small and $\alpha$ is small, then the number of good models being selected could be extremely low (possibly zero) since not enough variables may be selected from previous steps to build $m$ combinations. In general, we would want a large $m$ (so that $\alpha$ can eventually be chosen to be very small) and, remembering that $p^\star$ is the number of variables released from Algo. 1, a rule-of-thumb is to set $m = \binom{p^\star}{2}$ (or close to it) in order to explore the entire (or most of the) subspace of two-dimensional models generated by $p^\star$.

- $\alpha$: as discussed in the previous point, this parameter is related to the maximum number of models $m$. The larger $\alpha$, the more the variable space is explored. Ideally, we want to choose a small $\alpha$ since we would want to select good models (with extremely low error) and this is possible if $m$ is large enough. Generally good values for $\alpha$ are 0.01 or 0.05, implying that (roughly) 1% or 5% of the $m$ models are selected at each step.

The SWAG could be modified to perform tests to determine the best models across the different dimensions (e.g. equivalence tests), therefore removing the need to specify $\alpha$. This modification however is left for future research. A study on the sensitivity of the SWAG to these meta-parameters is given in App. D where results suggest that algorithm outputs are stable with respect to these choices.

## 2.2 SWAG Libraries and Rashomon Sets

While it must be stressed again that the SWAG does not aim to find a Rashomon set (but simply to address the Rashomon Effect in high-dimensional settings), it is reasonable to ask ourselves how the library of SWAG models $\widetilde{\mathcal{M}}$ (or $\widetilde{\mathcal{M}}_{\mathcal{R}}$ after post-processing) compare to the formal definition of a Rashomon set. For this reason, let us recall this definition which generally starts from a reference model $l^* \in \mathcal{L}$, based on which the Rashomon set $\mathcal{R}_\theta(l^*)$ is defined as:

$$\mathcal{R}_\theta(l^*) := \{l \in \mathcal{L} \mid D(l) \leq D(l^*) + \theta\},$$

where $D$ is a loss function (in our notation $\epsilon$) and $\theta > 0$ is a user-defined parameter which defines the margin within which model performance can be considered "almost-equivalent" (see e.g. Fisher et al., 2019). In this respect, the SWAG does not predefine a reference model $l^*$ or an "almost-equivalence" parameter $\theta$ but finds "almost-equivalent" good sparse models through the percentile $\alpha$ which is therefore not absolute (like $\theta$) but relative to the performance of all models evaluated in a given dimension $\hat{p}$. Following the definition of a Rashomon set, technically speaking, if we were to define a reference model $l^*$ then there would exist a $\theta$ for each dataset and model class such that the SWAG library constitutes a sparse subset of the respective Rashomon sets. However this is a post-hoc adaptation of the definition since the exploration of "almost-equally" performing models occurs differently between the SWAG and the formal definition of a Rashomon set. This being said, the SWAG can easily be modified to more directly search for models in the Rashomon set by selecting some sparse models whose loss (e.g. generalization error) is within a $\theta$-range of a pre-specified model $l^*$. Therefore, while these sets are indeed built on different criteria (and are thus

not directly comparable in terms of outputs), one could argue that the SWAG targets a sparse subset of the Rashomon set (assuming this subset exists) for a given model class $\mathcal{L}$ (see results in Sec. 3), thereby enhancing its interpretability for practitioners—particularly in light of the growing emphasis on "Explainable AI" (see e.g. Confalonieri et al., 2021, for a historical perspective). Moreover, under certain aspects, it can be considered more intuitive in the way models from a certain class are selected. For example, the SWAG does not require the pre-specification of a reference model $l^*$ and, in addition, the parameter defining the models in the SWAG set is a performance percentile rather than a pre-specified parameter $\theta$ related to the loss value $D$ (for which a reasonable range may sometimes be hard to determine *a priori*). However, as opposed to some currently available Rashomon set procedures (e.g. Xin et al., 2022; Qinyu Zhu et al., 2023; Zhong et al., 2024), the SWAG does not guarantee to find *all* models (but only some) with almost-equivalent performance for dimensions $1 \leq \hat{p} \leq p_{\max}$.

## 3 Empirical Results

We study the empirical performance of the SWAG with different model classes and on different datasets taken from the UCI Machine Learning Repository (see Dua & Graff, 2017), ArrayExpress (see Kolesnikov et al., 2015) and from the GitHub repository `https://github.com/ramhiser/datamicroarray`. More specifically, the chosen model classes are: (i) Lasso (logistic) (Friedman et al., 2010); (ii) Linear-Kernel Support Vector Machine (L-SVM) (see e.g., (Vapnik, 2013)); (iii) Radial-Kernel SVM (R-SVM) (Cortes & Vapnik, 1995); (iv) Random Forest (RF) (Breiman, 2001a). Compared to these models classes, the use of more complex models, such as neural networks, requires more thought on how to adapt their architecture as the number and combination of variables change throughout the SWAG (this is left for future work). To ensure a fair comparison, we run all the analyses with the `caret` R package (see e.g. Kuhn, 2008, for a review). All the hyper-parameters specific to each model class are set based on common/default choices and are discussed in App. C.

With regards to the datasets, they are the following: (i) MeterA (Gyamfi et al., 2018); (ii) LSVT (Tsanas et al., 2013); (iii) Ahus (Haakensen et al., 2016); (iv) Colon (Alon et al., 1999); (v) Leukemia (Golub et al., 1999). The choice of these datasets is based primarily on their high-dimensional nature ($n \ll p$) which is the setting that the SWAG was mainly conceived for. The details regarding these data and the choices for the SWAG meta-parameters are listed in Tab. 1: the choice of the SWAG meta-parameters is made in line with the rules of thumb described in Sec. 2.1 and with the intent of reducing overall computational time while

Table 1: For each dataset: # of instances per training and test set, # of variables and relative SWAG meta-parameter choices.

| Data | $n_{\text{train}}$ | $n_{\text{test}}$ | $p$ | $p_{\max}$ | $m$ | $\alpha$ |
|---|---|---|---|---|---|---|
| MeterA | 69 | 18 | 666 | 6 | 3984 | 0.05 |
| LSVT | 100 | 26 | 310 | 6 | 1014 | 0.05 |
| Ahus | 125 | 31 | 15,739 | 8 | 99080 | 0.01 |
| Colon | 50 | 12 | 2,000 | 4 | 7996 | 0.03 |
| Leukemia | 58 | 14 | 7,129 | 4 | 10160 | 0.01 |

ensuring a reasonable exploration of the variable space. More specifically, we choose $0.01 \leq \alpha \leq 0.05$ as a good compromise between fixing an $\alpha$ as small as possible (to select good models) and the possibility of exploring a considerable portion of the variable space. With $m$ depending on the computing time and power available, we choose $m$ in order to at least reasonably explore the subspace of two-dimensional models for all the model classes and datasets. The choice of $p_{\max}$ is made guided by the EPV rule discussed in van der Ploeg et al. (2014). As mentioned earlier, an extended study and discussion on the sensitivity of the SWAG results with respect to the choice of these meta-parameters can be found in App. D where, for example, it is also underlined how similar choices are needed for Rashomon sets regarding the choice of a reference model and "equivalence" parameter $\theta$. Finally, we apply 10-repeated 10-fold cross-validation (i.e. $r = 10$ and $k = 10$) as the error metric $\epsilon$ and choose $\delta = 0.01$ for the post-processing based on the previously described median rule. With the above in mind, we present some summary measures of the set of SWAG models $\widetilde{\mathcal{M}}$ such as their dimensions and their Jaccard diversity indices, allowing to understand how interpretable (sparse) and how flexible (diverse) the resulting SWAG models are (see App. E.3 and E.4 for details on number of SWAG models $|\widetilde{\mathcal{M}}|$ and Jaccard index). Tab. 2 shows the range of dimensions of the SWAG models ($|s_l|$), the median value for the pairwise Jaccard indices for all models in $\widetilde{\mathcal{M}}$ ($\text{med}_J$) as well as their range ($\text{range}_J$) for each dataset. It can be seen how the SWAG models are all extremely sparse (also as a result of $p_{\max}$) and

Table 2: Model class (main columns) and datasets (rows): (i) range of SWAG model dimensions, (ii) median Jaccard index and (iii) range of the Jaccard indices for the SWAG models.

| | Lasso | | | L-SVM | | | R-SVM | | | RF | | |
|---|---|---|---|---|---|---|---|---|---|---|---|---|
| | $|s_l|$ | $\mathrm{med}_J$ | $\mathrm{range}_J$ | $|s_l|$ | $\mathrm{med}_J$ | $\mathrm{range}_J$ | $|s_l|$ | $\mathrm{med}_J$ | $\mathrm{range}_J$ | $|s_l|$ | $\mathrm{med}_J$ | $\mathrm{range}_J$ |
| MeterA | [4, 6] | 0.38 | [0.09, 0.83] | [2, 6] | 0.16 | [0.00, 0.83] | [4, 6] | 0.20 | [0.09, 0.83] | [4, 6] | 0.33 | [0.09, 0.83] |
| LSVT | [4, 6] | 0.15 | [0.20, 0.83] | [5, 6] | 0.09 | [0.00, 0.83] | [4, 6] | 0.38 | [0.33, 0.83] | [3, 6] | 0.25 | [0.00, 0.83] |
| Ahus | [6, 8] | 0.15 | [0.00, 0.88] | [7, 8] | 0.23 | [0.00, 0.88] | [5, 8] | 0.23 | [0.00, 0.75] | [5, 8] | 0.25 | [0.00, 0.75] |
| Colon | [3, 4] | 0.14 | [0.00, 0.75] | [3, 4] | 0.33 | [0.00, 0.75] | [3, 4] | 0.14 | [0.00, 0.75] | [3, 4] | 0.40 | [0.33, 0.75] |
| Leukemia | [2, 3] | 0.00 | [0.00, 0.67] | [2, 3] | 0.20 | [0.00, 0.67] | [2, 3] | 0.00 | [0.00, 0.67] | 3 | 0.50 | [0.20, 0.50] |

that at least half of the Jaccard indices for all model classes and datasets are below or equal to 0.5, indicating that there is a reasonable or high level of diversity in the SWAG models which is essential for user-flexibility as well as interpretability of latent structures.

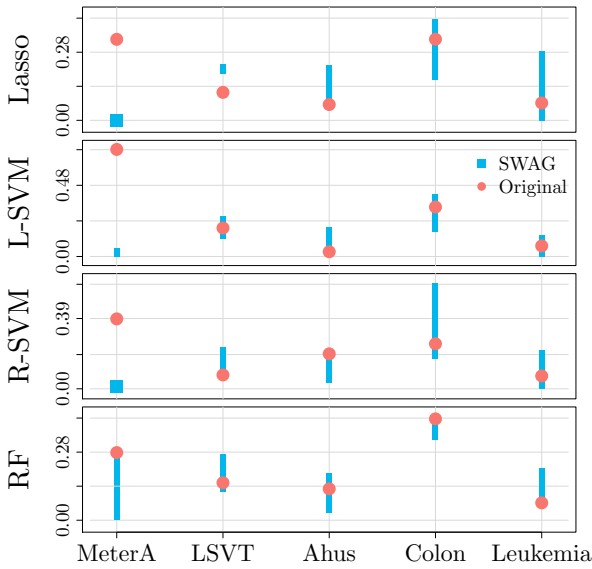

Figure 1: SWAG test error range $\epsilon_{\text{test}}$ (blue rectangles) and original model test error (red dots). The blue *squares* imply that the SWAG models all have the same test error.

Indeed, with respect to user-flexibility, variable diversity can be important for practitioners who need similarly good models with variables that are less costly/invasive to collect and measure, or that can be used when different institutions do not collect the same variables (see e.g. Kissel & Mentch, 2024). However, an important aspect to underline is that, while the goal of the SWAG is to deliver the latter two advantages, these seem to come with a limited cost in terms of prediction error with respect to the original reference models, thereby suggesting it is capable of finding *sparse subsets of the Rashomon set* (if these exist). Indeed, Fig. 1 represents the test error, denoted as $\epsilon_{test}$, of the original models (i.e. models trained using all variables) represented by the red dots, and of the respective SWAG models, represented by a blue rectangle proportional to the range of test errors (a similar figure is available for the training error $\epsilon_{train}$ in App. E.2). It can be seen how in the majority of cases the SWAG models have close or comparable (if not sometimes better) prediction accuracy with respect to their original versions, with homogeneous and exact accuracy when using Lasso and R-SVM on the Meter A data for example. This range of performance of SWAG models is expected and in line, for example, with the definition

of Rashomon sets which are built on "almost-equivalent" models within a $\theta$-range of the reference one. Indeed, in our case, the prediction performance is overall preserved while selecting extremely sparse models (maximum dimension 8) while the Lasso (the only sparse alternative considered) selects between 10 and 26 variables. Moreover, the user can arbitrarily choose to trim these distributions further by restricting their final selection to the best models in the SWAG library.

Further extensive experiments are provided in App. E supporting the stability of the SWAG across (i) meta-parameters, (ii) datasets and (iii) model classes. Indeed, conclusions similar to those presented above hold even when using existing (step-wise) variable selection approaches as possible good reference models as well as across different meta-parameter and model-class choices (see e.g. App. A and D). Moreover, a simulation setting based on latent structures is provided in App. E.5 suggesting that the SWAG can uncover latent connections that justify the existence of almost-equivalent models (such as Rashomon sets). This being said, given the fact that prediction accuracy and stability is generally preserved with the SWAG models,

let us investigate the advantages of the SWAG in terms of interpretability of latent structures (along with user-flexibility which was observed through the Jaccard index above). To do so we consider building SWAG networks, based on the final selected set of models, in which the size of the nodes are proportional to the presence of a variable in SWAG models while the thickness of edges is proportional to times the variables are in the same SWAG model. We briefly study and interpret the latent structures highlighted by these networks for two of the considered datasets (more information in App. E.6).

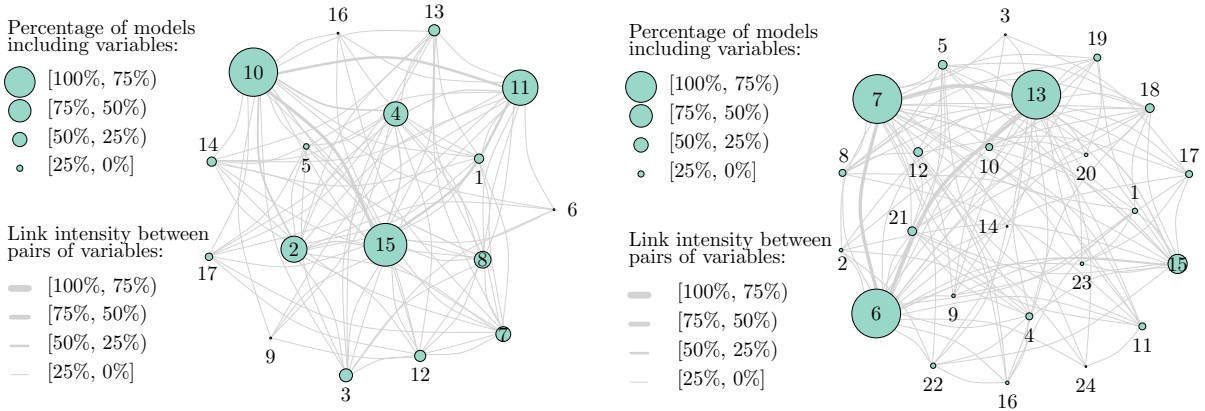

Figure 2: Network of variable importance and pairwise link-intensity in $\widetilde{\mathcal{M}}$ for Meter A dataset using the Lasso-based SWAG.

Figure 3: Network of variable importance and pairwise link-intensity in $\widetilde{\mathcal{M}}$ for LSVT dataset using the Radial-Kernel SVM SWAG.

**Ultrasonic Flowmeters** The Meter A data is analyzed in Gyamfi et al. (2018) and collects measurements on ultrasonic flowmeter diagnostics. Achieving good diagnostics regarding the health of a flowmeter is of extreme importance for condition-based maintenance in many industrial sectors such as the oil and gas industry since incorrect measurement can entail considerable economic and material losses (see e.g. TUV-NEL, 2012). In this data the goal is to classify the health of a meter into two classes: "Healthy" (Class 1) or "Installation effects" (Class 2). Given that the variables are measurements of physical nature, we decide to consider all first-order interactions which finally delivers a total variable size of $p = 666$ (36 original variables plus $\binom{36}{2} = 630$ interactions). Indeed, we do this since these interaction effects could not be adequately captured by models that only contain the variables of interest on their own (see e.g. Jaccard & Turrisi, 2003). Using the results from Lasso-based SWAG, the network in Fig. 2 would suggest that in order to understand the mechanics and perform diagnostics for this ultrasonic flowmeter, among the 666 variables, an engineer could focus their attention on variables labeled 10, 11 and 15 which correspond to the interaction (i) between flatness ratio and gain as well as (ii) between the speed of sound and gain at the first end (of the fifth path). For instance, the interaction between flatness ratio and gain is significant since, if the flatness ratio is low, it indicates possible disturbances in the flow that could affect the gain measurements, leading to potential inaccuracies in flow estimation. Moreover, combining the internal speed of sound with gains can help to detect anomalies that individual measurements might miss. This relationship suggests that efficiency in signal transmission could degrade over time and impact various paths differently. These variables are all linked to characteristics such as structural integrity, alignment quality and environmental conditions, suggesting that these could be the latent variables affecting the health of the flowmeter (Lynnworth & Liu, 2006).

**LSVT** The voice rehabilitation dataset was analyzed in Tsanas et al. (2013) in order to assess the effectiveness of a computer program called "Lee Silverman voice treatment (LSVT) Companion" which allows patients with Parkinson's disease (PD) to independently progress through a rehabilitative treatment session. Taking data on 126 samples from 14 patients who followed the latter treatment, 310 dysphonia measures were taken on each of them (plus information on sex and age of the patients) and used to understand if they could correctly predict whether the patients' voices were "acceptable" or "unacceptable" after this treatment. There

is also scientific interest in determining the variables (and combinations thereof) that most contribute to the definition, in this case, of a Parkinson's speech treatment as being acceptable or not. Also in this case, the SWAG models (based on the R-SVM) can be arranged into a network to allow for interpretation as seen in Fig. 3. Based on the SWAG network, researchers interested in improving speech treatment should focus on variables 6, 7 and 13 which correspond to the $2^{nd}$ and $3^{rd}$ Mel-Frequency Cepstral Coefficients (MFCC) and to the entropy with base-4 logarithmic coefficients (as well as the interactions between these three variables as highlighted by their frequent presence in the same SWAG models). These variables indirectly measure latent characteristics such as voice quality and cognitive load: (i) fluctuations in MFCC can reflect changes in the vocal tract configuration, corresponding to mechanical or motor control difficulties in speech production in PD patients while (ii) entropy values can indirectly suggest the cognitive effort involved in speech production which, in patients with PD, can lead to reduced speech variability and difficulties in modulation, impacting treatment effectiveness (Dao et al., 2022; Zhang et al., 2023; Shen et al., 2024).

**Network Interpretation**  These networks allow to look at variable importance in a different manner. Indeed, differently from current feature importance and attribution techniques (see e.g. Laberge et al., 2023), these networks allow to understand variable importance in conjunction with all other variables as well as to preserve the possibility of changing variable effects (attributions) according to the other variables considered in the network. More in detail, the co–occurrence of variables provides a structural view of how they interact across the SWAG library. For example, the measures of frequency (marginal importance) and degree (co-occurence) capture complementary aspects of structural importance within the model space: they respectively quantify how often a variable appears and how broadly it co-occurs with others. Other network measures such as betweenness centrality highlight variables that act as bridges between otherwise weakly connected regions of the network, and eigen centrality emphasizes membership in the influential core of variables. Indeed, limiting ourselves to Fig. 2 as an example, while it is often the case that certain variables appear to be essential (e.g. variable 10 since all other variables are connected to it), there are some more frequently present variables that do not need to be included in a model to achieve comparable predictive accuracy (e.g. variable 15 in Fig. 2) whereas variables 2 and 5 display exceptionally high betweenness scores despite a moderate or low frequency, implying that they serve as a bridge between clusters of predictors that are otherwise rarely connected. This bridging function may indicate that these variables capture a general effect or contextual factor that connects distinct modeling scenarios. In this sense, the SWAG networks therefore allow the user to understand how other combinations of variables can replace an "important" variable. This way of interpreting the results allows to not only look at single variable importance, but also at *variable combination importance*. This interpretation emphasizes how network centrality measures can highlight not only which predictors are important, but also *how* they contribute, i.e. either by stabilizing model structure, linking distinct predictor groups, or capturing specialized effects. This can have many practical implications, including the possibility for the user to employ other variables to achieve predictive accuracy when, for example, the measurements for other variables are missing. The specific interpretation for these networks through their networks can be found in App. E.6.

## 4   Conclusions

The proposed algorithm does not directly fit in the standard procedures to build Rashomon sets since these are commonly found starting from a reference model and explore all model dimensions. More specifically, without a reference model and by fixing a model class, the SWAG assumes the existence of good predictive sparse models within this class and aims to find (some of) them. By doing so, we can see that the SWAG produces many of the main results of the Rashomon Effect (Rudin et al., 2024), in particular: (i) the selection of many sparse-yet-accurate models (on par with their original counterparts applied to all variables) as shown by Fig. 1; (ii) flexibility to provide users with diverse and sparse models that are interchangeable in terms of predictive performance as highlighted by Tab. 2; (iii) evaluation of prediction uncertainty through SWAG error (or accuracy) ranges, in particular model-uncertainty (see again Fig. 1); (iv) stable variable importance metrics that can be interpreted also in terms of connections with other variables (see Fig. 2 and 3). Moreover, as shown in Sec. 3, all this is achieved through any model class of choice. Indeed, the possibility of studying the role of variables in a network (i.e. the change of effects when combined with other variables) can also be a tool to address the *replication crisis* occurring in many domains. In this sense, finding the best single

model in each study does not allow to assess how generalizable the successive interpretations can be, whereas sets of models can stabilize interpretations by verifying which variables have stable effects in the network. In particular, these properties directly address some of the requirements of the PCS framework of Yu & Kumbier (2020), specifically the model-perturbation that ensures that the final model(s) are more generalizable.

Given its heuristic nature, the proposed algorithm can be easily adapted for different user preferences, including the selection of sparse model sets based on domain-specific utility (cost) functions. Future directions of research for this procedure include the theoretical study of its properties when in the presence of sparse latent representations, including its ability to select models built with weakly correlated variables which are linked to distinct underlying latent functions. Moreover, the structure of the SWAG can possibly allow to construct inferential tools to determine the significance of the model set $\widetilde{\mathcal{M}}$, specifically if there are indeed important variables/models or if the set contains randomly selected models due to spurious correlations and/or to the absence of significant predictors. Using the network representations of the SWAG models, one could employ network statistics to overcome the problem of testing each model in the set individually, thereby delivering the first statistical tools for testing sets of models (e.g. Rashomon sets).

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

# A    Comparison to Stepwise Single-Model and Set-of-Models Methods

We firstly give a brief overview of stepwise variable selection methods and then discuss existing procedures that instead select sets of models. Later, we provide some comparisons between them and the SWAG framework. Therefore, let us start by recalling the main strategies in forward or backward selection as commonly applied to linear and logistic regression model classes. Probably the most well-known procedure, and still one of the most employed by practitioners, is plain *forward selection*. It starts from an empty model and iteratively adds the variable that results in the greatest improvement of a criterion e.g., the *Akaike Information Criterion* (AIC; Akaike, 1973) or the *Bayesian Information Criterion* (BIC; Schwarz, 1978). Historically, the roots of stepwise selection can be traced to Efroymson (1960), and various stepwise approaches remain standard for their simplicity, although they usually are not particularly suited for high-dimensional problems. To tackle this setting, another group of methods leverages *screening* which can then be combined with forward selection or any other feature selection procedure (similarly to SWAG). For example, in ultrahigh-dimensional settings, *Sure Independent Screening* (SIS) proposed by Fan & Lv (2008) and Fan et al. (2009) ranks variables by a measure of marginal association, drastically reducing the dimensionality. Its iterative counterpart, ISIS, refines this process by allowing repeated screening steps combined with penalized methods such as Lasso (Tibshirani, 1996), SCAD (Fan & Li, 2001), or MCP (Zhang, 2010). A further popular method is *Recursive Feature Elimination* (RFE), introduced by Guyon et al. (2002). RFE begins with the full set of variables and then: (i) trains a model within the class of choice e.g., Random Forests, (ii) computes variable-importance metrics, and (iii) eliminates the least important variables, iterating until a specified subset size is reached.

All the above methods (and, to the best of our knowledge, the vast majority of existing forward/backward and penalized variable selection algorithms) aim to find a single, final subset of features i.e., a single model. With SWAG, we build on the underlying idea of forward selection (i.e. growing from smaller to larger subsets of variables) but adapt it to identify multiple sparse and accurate models rather than a single best model. In essence, SWAG is a model-agnostic wrapper that uses any estimator of the test error metric, e.g. cross-validation (CV), to guide each enlargement of the variable set. It screens variables one at a time initially, then expands only those subsets that have already demonstrated strong predictive performance, and it does so for any chosen model class, whether linear/logistic regression, kernel SVMs, Random Forests or possibly even more complex learners. Moreover, one of the main reasons for running SWAG is to study the so-called Rashomon Effect (Breiman, 2001b; Fisher et al., 2019) where various models can deliver similarly good performance on a given dataset. Instead of discarding the "almost-as-good" alternatives, the SWAG keeps them therefore letting users explore the diversity of good solutions, check for potential sparse latent structures and build variable-importance networks to interpret effects. In comparing SWAG with existing methods, one can note that it preserves the forward-growing nature of classic stepwise selection but goes beyond it by exploring multiple subsets in parallel at each dimension rather than following a single path. This allows SWAG to present a library of equally good models rather than a single final model. Although nothing impedes from modifying SWAG in various ways, currently it does not rely only on marginal metrics for screening (differently from screening-based methods such as SIS and ISIS) but directly uses the test error $\epsilon$ which, after the first step of the procedure, accounts for conditional effects given other variables. Compared to RFE, SWAG pursues a forward approach rather than a backward one, which tends to be more computationally tractable when the number of variables is large. Also, RFE methods typically require a sensitivity metric specific to the chosen class of models and still aim for a single best subset of variables, whereas SWAG by construction accommodates different classes of models and retains many good subsets of variables. Lastly, it is interesting to compare SWAG with a variable selection method that also delivers a set of models i.e., the *random subspace method* (RSM) of Ho (1998). RSM is an *ensemble* approach in which variable subsets are chosen at random and used to train weak models. On the other hand, SWAG focuses on identifying good (strong) sparse models through a deliberate search guided by the error metric $\epsilon$. Because RSM is inherently random, interpreting variable importance can be more challenging, whereas SWAG keeps track of which variables produce good performance and thus highlights how features co-occur across good models. Of course, this comes at a computational cost as SWAG by construction, is more computationally intensive than the classical RSM. A recent alternative very similar in spirit to the SWAG is the MPS of Kissel & Mentch (2024) which builds "forking-paths" from each of the pre-screened variables and evaluates (in a forward stepwise manner) which variables should be added to each of the previously selected variables.

By using a stability selection procedure in each step, the MPS builds a set of models of size $p_{max}$. Compared to the latter, the SWAG is less greedy in terms of nested structures since it allows to drop certain previous variable combinations if they do not improve the error metric in the following dimensions. Moreover, it does not only consider models of dimension $p_{max}$ but also all smaller models with similarly good performance (although we think that the MPS could theoretically be adapted to consider this case as well). Based on these characteristics, while being more computationally burdensome, the SWAG would intuitively provide more diverse models in terms of size and variable combinations which can be an important aspect for practitioners.

To provide a practical comparison, we run the previously discussed methods on the same datasets and with the same model classes as those shown in Sec. 3. More in detail, in Tab. 3 we compare SWAG with: (i) AIC forward selection combined with the SIS screening procedure (*SIS + forwardAIC*); (ii) the iterative SIS coupled with the default MCP penalization method (*ISIS MCP*);(iii) the RFE with Random Forests as the model class (*RFE RF*) and (iv) the RSM for each model class. These competitors encompass selection approaches of different nature, including penalized-based methods (e.g. Lasso, ISIS) and permutation importance based methods (RFE based on RF with MDA measures), as well as the RSM, which instead selects a set of models. We underline that we coupled classic forward selection with a screening procedure because in itself it is not suited for the high-dimensional datasets used in our work. This is because running a logistic regression requires a number of observations larger or equal to the number of variables, which is not the case for all our datasets. Throughout this analysis, we have made an effort to perform the most fair comparison possible. In particular, we have used 10-fold CV for hyperparameter tuning and we input our choice of $p_{max}$ to both *RFE RF* and *RSM* for each dataset, to speed-up and facilitate the search in the space of models of the considered competitors. For RSM, we followed the guidelines of Tian & Feng (2021), using 200 base models for medium-dimensional datasets (i.e., MeterA, LSVT, and Colon) and 500 for the high-dimensional Ahus and Leukemia datasets. The main result of the applied analysis for single-model methods (see Tab. 3) is that, for all the datasets considered, the test error $\epsilon_{test}$ of each competitor lies within the test error range of the SWAG models (or even above the maximum value in some cases) for the relevant model class i.e., Lasso-logistic for *SIS + forwardAIC* and *ISIS MCP* and Random Forests (RF) for *RFE RF*. These results provide further evidence that SWAG searches the space of models in a comparable way to other forward/backward feature selection methods but, as opposed to the latter, outputs a library of almost-equally-good sparse models rather than a single one. This result also complements the one presented in the paper (see Fig. 1) where the test error of each full model, trained with all the variables, is always either within the SWAG range or sometimes even higher than the worst test error achieved by a SWAG model (except for one case). For the set-of-models method comparison, we note (see Tab. 3) that the RSM yields wider test error ranges than SWAG across all model classes, including lower test errors for the lower bound of the ranges. This is expected, as the random exploration of the model space in RSM generates a highly diverse collection of models[2] whose predictive performance varies substantially, making the ensemble less homogeneous and thus less reflective of the Rashomon Effect, compared to SWAG sets of models. Indeed, the RSM does not attempt to address the Rashomon Effect since it does not aim to find equivalently well-performing models, but simply a set of models to be then used in an ensemble approach. This can be observed in the values of the median test performance, as well as the upper bounds of performance ranges, which suggest that the RSM models have errors distributed across the broader range of errors while the SWAG errors are more concentrated towards the lower end of the range (where the Rashomon Effect takes place). The only exception to this rule is the Colon dataset where RSM has a a better overall performance, although the SWAG models are still within range. Moreover, all models for RSM are of the same dimensions (i.e. $p_{max}$) which do not make the results directly comparable to those of the SWAG which are based on models of smaller and different dimensions. In fact, since $p_{max}$ is generally small (compared to $p$), it would not be surprising to randomly find better performing models of that dimension compared to those contained in the SWAG. In addition, the RSM would not allow to study variable importance or co-occurence patterns (through the network) and, despite high Jaccard diversity, would always require to find at least $p_{max}$ variables to obtain predictions (as opposed to the SWAG which provides sparser alternatives with similar accuracy to the best in the set).

---

[2]The value of the median Jaccard index is 0 for every model class and for each dataset, with a maximum index ranging between 0.06 and 0.35 across model classes and datasets. This is mainly due to the high-dimensional nature of the datasets where the probability of randomly picking overlapping combinations of dimension $p_{max}$ is lower.

| Method | MeterA | | LSVT | | Ahus | | Colon | | Leukemia | |
|---|---|---|---|---|---|---|---|---|---|---|
| | Median | Range | Median | Range | Median | Range | Median | Range | Median | Range |
| SWAG range Lasso | **0.000** | **[0.000, 0.000]** | **0.192** | **[0.192, 0.231]** | **0.129** | **[0.065, 0.226]** | **0.417** | **[0.167, 0.417]** | **0.071** | **[0.000, 0.286]** |
| *RSM range Lasso* | 0.333 | [0.000, 0.778] | 0.269 | [0.115, 0.692] | 0.419 | [0.065, 0.645] | 0.333 | [0.083, 0.667] | 0.357 | [0.071, 0.571] |
| *SIS + forwardAIC* | — | 0.111 | — | 0.192 | — | 0.129 | — | 0.417 | — | 0.143 |
| *ISIS MCP* | — | 0.333 | — | 0.231 | — | 0.065 | — | 0.250 | — | 0.143 |
| SWAG range RF | **0.083** | **[0.000, 0.278]** | **0.154** | **[0.115, 0.269]** | **0.065** | **[0.032, 0.194]** | **0.417** | **[0.333, 0.417]** | **0.143** | **[0.071, 0.214]** |
| *RSM range RF* | 0.333 | [0.056, 0.889] | 0.231 | [0.077, 0.462] | 0.419 | [0.097, 0.710] | 0.333 | [0.083, 0.583] | 0.286 | [0.000, 0.714] |
| *RFE RF* | — | 0.000 | — | 0.154 | — | 0.129 | — | 0.417 | — | 0.071 |
| SWAG range L-SVM | **0.000** | **[0.000, 0.056]** | **0.192** | **[0.115, 0.269]** | **0.129** | **[0.065, 0.194]** | **0.417** | **[0.167, 0.417]** | **0.071** | **[0.000, 0.143]** |
| *RSM range L-SVM* | 0.389 | [0.000, 0.500] | 0.308 | [0.115, 0.423] | 0.452 | [0.097, 0.645] | 0.333 | [0.083, 0.583] | 0.357 | [0.071, 0.643] |
| SWAG range R-SVM | **0.015** | **[0.007, 0.016]** | **0.115** | **[0.077, 0.231]** | **0.065** | **[0.032, 0.194]** | **0.333** | **[0.167, 0.583]** | **0.071** | **[0.000, 0.214]** |
| *RSM range R-SVM* | 0.389 | [0.000, 0.833] | 0.269 | [0.077, 0.538] | 0.452 | [0.065, 0.677] | 0.333 | [0.083, 0.583] | 0.357 | [0.071, 0.571] |

Table 3: Comparison of test errors $\epsilon_{\text{test}}$ across methods, with per-dataset subcolumns for the median and range. Current entries populate the *range* subcolumns using available results; median values are left blank (—) to be completed. For single-model baselines (*SIS + forwardAIC*, *ISIS MCP*, *RFE RF*) the existing point estimates are shown in the *range* subcolumns. Datasets: MeterA, LSVT, Ahus, Colon, Leukemia.

To summarize, SWAG brings two main benefits compared to existing methods: (i) by using any test error metric, it can wrap around any model class without requiring specialized ranking metrics (each tied to a specific class of models); (ii) it finds multiple *sparse* and *diverse* almost-equally-good (distinct) solutions at each subset size to address the Rashomon Effect. In terms of pros, empirical performance this far shows that SWAG can match or exceed the accuracy of a (regularized) model fit on the whole data while selecting very few variables, and it allows for new avenues of interpretation (e.g. through a network of co-occurrences among variables in the best subsets). On the cons side, SWAG's forward-greedy expansion can be computationally demanding if the number of variables is extremely large or if the error metric $\epsilon$ is burdensome to compute, although parallelization can mitigate these burdens together with a smarter choice of the error metric if available (e.g. AIC or BIC). In this perspective, since the parameter $p_{\max}$ can work as a regularization parameter, it may be possible to simply evaluate the models based on their training error thereby avoiding major computational costs coming from cross-validations (to be investigated in future work). Moreover, it is still not known if and under what conditions the SWAG guarantees finding *every possible* almost-equally good model in the considered dimensions, though in practice it has proven to robustly capture diverse and accurate solutions for different model classes and datasets. To conclude, the SWAG fits into the forward selection tradition by however extending its scope: instead of identifying a single final subset it finds multiple sparse subsets with good performance and delivering advantages common when taking into account the Rashomon Effect.

## B   Computational Complexity

We firstly underline that the premise of the SWAG is to deliver a set of good sparse models for high-dimensional problems to be employed at the user's discretion for interpretations, predictions and decisions. Therefore it does not consider problems where the user needs to rapidly re-estimate or re-test new models and consequently assumes that the user is not constrained by time to obtain the SWAG outputs. This being said, the SWAG is a wrapper algorithm and therefore part of its complexity depends on the complexity in fitting the model class run within it. For this reason, let $C(n, \hat{p})$ represent the complexity of fitting a model $l \in \mathcal{L}$ to a dataset with $n$ samples and $\hat{p}$ variables. Based on this, the first step of the SWAG evaluates each variable individually, therefore the complexity of this first step is $O(p\, C(n, 1))$. The general step iteratively fits models with $2 \le \hat{p} \le p_{\max}$ variables where, for each dimension $\hat{p}$, the number of times the model is fitted is $m$ with consequent complexity $O(m\, C(n, \hat{p}))$. Since the latter step iterates up to $p_{\max}$, the worst-case complexity of the general step is $O((p_{\max} - 1)\, m\, C(n, p_{\max})) = O(p_{\max}\, m\, C(n, p_{\max}))$. Therefore, combining the two steps of the SWAG delivers an overall worst-case complexity of $O(p\, C(n, 1) + p_{\max}\, m\, C(n, p_{\max}))$. In addition, if one uses $r$-repeated $k$-fold cross-validation to obtain the test error metric $\epsilon$, then this worst-case complexity grows to $O(r\, k\, (p\, C(n, 1) + p_{\max}\, m\, C(n, p_{\max})))$. As a consequence, since $p$ is fixed by the data and $p_{\max}$ is fixed to be small by the user for sparsity/interpretation reasons, the computational complexity is driven mainly by the choice of $m$ (i.e. the number of models to evaluate at each step) and, in particular,

by the test error metric chosen. Indeed, the overall complexity of the SWAG can be increased $(r\,k)$-fold if choosing $r$-repeated $k$-fold cross-validation but can be considerably reduced if one is able to employ other more efficient metrics. As an example, if using a generalized linear model with complexity $O(n\hat{p}^2 + \hat{p}^3)$ and using AIC or BIC as the error metric $\epsilon$, then the worst-case complexity becomes $O\left(np + mnp_{\max}^3 + mp_{\max}^4\right)$: this obviously consists in a considerable decrease compared to the general worst-case complexity, as also shown the computational runtimes further on. One could even further reduce computational complexity by directly using training error metrics since the parameter $p_{\max}$ could already be considered as a regularizer to limit over-fitting (see Efron et al., 2004, and discussions in Sec. 2.1). In addition to $m$ however, in specific cases the parameter $\alpha$ can also play a role for complexity with respect to the number of variables $p$. Indeed, $\alpha$ determines the set of screened variables $\mathcal{S}^*$ whose size is $p^*$. In the general step the SWAG samples $m$ possible combinations out of the $\binom{p^*}{\hat{p}}$ available: this total number of available combinations may be smaller than $m$ due to a potentially small $p^*$, in which case the SWAG performs an exhaustive search and evaluation of all possible variable combinations for dimension $\hat{p}$ (i.e. best subset selection). If this occurs, then this will be the case also in larger dimensions up to $p_{\max}$ where the number of possible combinations will actually decrease with the dimension. As a result, the complexity coming from $m$ will be reduced to $O\left(\binom{p^*}{\hat{p}}\right)$ for all $\{2 \le \hat{p} \le p_{\max} \,|\, \binom{p^*}{\hat{p}} < m\}$.

### B.1  Example Runtime: Leukemia Dataset

We ran the SWAG on an Apple 8-core M1 pro 16GB RAM on the Leukemia dataset which has $n = 58$ training samples and $p = 7129$ variables with $m = 10160$, $\alpha = 0.01$ and $p_{\max} = 4$. Using 10-repeated 10-fold cross-validation to estimate the error metric $\epsilon$, we get the results in Tab. 4 (rounded to the closest minute). These times will obviously be longer when the number of samples $n$ increases, especially when employing model classes that do not scale too well with $n$. These runtimes can however be significantly reduced through vectorized operations and, in particular, parallelization (when possible) since we can fit and evaluate multiple models for each dimension $\hat{p}$ at the same time. Also, as mentioned earlier, the direct use of the training error would speed-up computations drastically, as is done by existing approaches to find Rashomon sets for example.

|      | Lasso | L-SVM | R-SVM | RF |
|------|-------|-------|-------|-----|
| SWAG | 50 min | 22 min | 153 min | 82 min |
| RSM  | 2.95 min | 0.45 min | 5.36 min | 3.04 min |

Table 4: Comparison of runtimes (in minutes) between SWAG and RSM on the Leukemia dataset across different model classes.

For reference, Tab. 4 also reports the runtimes of RSM (see section A for details on the choice of the hyperparameters) on the same dataset. As expected, RSM is considerably faster, since it performs a random search for much fewer models (500 vs $\approx 40,000$ of the SWAG) and does not require the estimation of any error metric. This computational complexity is further highlighted when comparing the SWAG runtimes with those of the single-model stepwise selection procedures considered in Appendix A: *SIS + forwardAIC* runs in 1.11 seconds; *ISIS MCP* in 24.82 seconds; *RFE RF* in 37.61 seconds. However, note that the use of a more computationally efficient error metric estimator can drastically reduce SWAG runtimes and make them comparable to RSM and single-model ones. For example, we can use a simple *logistic regression* in the SWAG and the AIC as the error metric $\epsilon$. Running it on a MacBook Pro with 1.4 GHz Quad-Core Intel Core i5 processor and 8 GB RAM, the SWAG procedure takes **0.32 min (roughly 20 seconds)**. Moreover, as a side note, this does not appear to affect the test error performance of the resulting SWAG models compared to the model classes and cross-validation approaches used for the other methods. In fact, the range of test errors for the SWAG models using this approach is $[0, 0.214]$ and the model dimension range is $[2, 4]$ therefore delivering results in line with the other model classes for this dataset. This suggests that if an efficient estimator of the test error is available, runtimes can be considerably reduced without significantly affecting the results.

## C  Hyper-parameter Choice

In this work, we make use of different model classes that we use either in their original versions or within the SWAG. For this reason, we make use of the same hyper-parameter selection rules in all settings, relying on the common/default choices (as per the `caret` R package):

1. **Lasso-logistic**[3]: the data is pre-processed (i.e., centered and rescaled) and a grid of 5 penalty terms ($\lambda$) for the Lasso (logistic) is used with $\lambda \in [0, \lambda_{max}]$ where $\lambda_{max} = \{0.05, 0.1\}$ depending on $p_{\max}$.

2. **Linear-** and **Radial-Kernel SVM**: in both cases, the data is pre-processed (i.e., centered and rescaled) and the Linear-Kernel SVM has a cost parameter $c = 1$ (default value), while the cost for Radial-Kernel SVM is automatically selected based on a grid of 5 values (i.e., `tuneLength = 5`);

3. **Random Forest**: the default value for the number of trees is kept (i.e., `ntree = 500`) as well as the common choice for the number of randomly sampled variables at each split (i.e., `mtry=` $\sqrt{p}$ for the original version and `mtry` $= \sqrt{\hat{p}}$ for the SWAG; see Genuer et al. (2010) for a detailed discussion.

## D  Sensitivity Analysis

We briefly discuss how the SWAG compares to Rashomon sets in the choice of their respective parameters (which was also mentioned in Sec. 2.2). Indeed, recalling the definition of Rashomon sets, the "performance" of the Rashomon set depends on the choice of a (good) reference model $l^*$ and the parameter $\theta$ which may not always be easy to determine according to the choice of the loss function $D$ (e.g. it may be difficult to have *a priori* knowledge of the potential range of loss values for a specific problem). Similarly, the performance of the Rashomon set can vary depending on choice of the model class $\mathcal{L}$ and across different datasets. Therefore, similarly to Rashomon sets, the choice of the meta-parameters is up to the user's discretion based on their needs. In particular, given that the parameter $p_{\max}$ is not chosen for performance but for user-preference on model dimensions, the main parameters to be chosen are $m$ and $\alpha$ (and eventually post-processing parameters, e.g. $\delta$). The parameter $\alpha$ plays a similar role to the parameter $\theta$ in Rashomon sets, except that it is not an absolute value but relative to the set of $m$ models evaluated in a certain dimension. In some way, the choice of $\alpha$ could be considered more intuitive with respect to $\theta$ since the former represents the proportion of best performing models we want to use in the algorithm, whereas the choice of $\theta$ relies on the possible (unknown) range of losses with respect to a model class and dataset. Therefore, the "additional" parameter with respect to Rashomon sets would be the number of models to evaluate in each step of the SWAG, i.e. $m$.

With the above in mind, we perform some sensitivity analyses to understand how these parameters can affect the SWAG results in terms of test error ($\epsilon_{\text{test}}$) and SWAG model dimensions ($|s_l|$). Given that $p_{\max}$ is determined by the user based on their interpretability requirements, as mentioned earlier, we focus on the impact of the parameters $m$ (number of models considered for each dimension) and $\alpha$ (the percentile to determine the selected models for each dimension). We firstly run the SWAG for different model classes and different combinations of meta-parameters $\alpha$ and $m$ for the Leukemia dataset: the results are presented in Tab. 5 and 6. We see that as the parameter $\alpha$ increases, so does the average test-error and its upper range, which is to be expected: by increasing the proportion of selected models we allow "worse-performing" models to be considered in the SWAG therefore increasing the upper bound (just like increasing $\theta$ for Rashomon sets). Across different values of $m$ we generally also see an increase in the upper range of test errors as we increase $m$: nevertheless for a fixed value of $\alpha$, the choice of $m$ does not appear to considerably affect the range of SWAG model dimensions $|s_l|$ or the average test-error performance $mean(\epsilon_{test})$. Indeed, as we would expect, increasing $m$ allows to explore more variable combinations and, by doing so, identifies more well-performing models as well as more "worse-performing" ones. Similar conclusions are made when focusing on the Meter A data using the Linear-Kernel SVM: see Tab. 7. All things being equal, these intuitive results suggest to pick a large $m$ (conditional on computational time and resources) and then let the user pick the best performing ones within the final SWAG library. Nevertheless, as seen in these sensitivity results, similarly good-performing models are found even with smaller values of $m$. In fact, the issue of consistent performance

---

[3]In this particular case, Algo. 1 is implemented with the `caret` package using classical (non-penalized) logistic regression

of the SWAG in some way relies mainly on whether the SWAG is able to find some good performing models independently from these parameters, and the means and ranges of test errors across all model classes suggest that this is generally the case.

| | $m = 5,000$ | | | $m = 10,000$ | | |
|---|---|---|---|---|---|---|
| | $\alpha = 0.1$ | $\alpha = 0.05$ | $\alpha = 0.01$ | $\alpha = 0.1$ | $\alpha = 0.05$ | $\alpha = 0.01$ |
| | | | Lasso-Logistic | | | |
| $\epsilon_{\text{test}}$ | $[0, 0.286]$ | $[0, 0.286]$ | $[0, 0.286]$ | $[0, 0.357]$ | $[0, 0.357]$ | $[0, 0.286]$ |
| $\text{mean}(\epsilon_{\text{test}})$ | 0.134 | 0.109 | 0.059 | 0.110 | 0.100 | 0.075 |
| $|s_l|$ | $[2, 4]$ | $[2, 4]$ | $[2, 4]$ | $[2, 4]$ | $[2, 4]$ | $[2, 4]$ |
| | | | Linear SVM | | | |
| $\epsilon_{\text{test}}$ | $[0, 0.357]$ | $[0, 0.214]$ | $[0, 0.214]$ | $[0, 0.357]$ | $[0, 0.357]$ | $[0, 0.142$ |
| $\text{mean}(\epsilon_{\text{test}})$ | 0.092 | 0.075 | 0.063 | 0.115 | 0.083 | 0.063 |
| $|s_l|$ | $[2, 4]$ | $[2, 4]$ | $[2, 4]$ | $[2, 4]$ | $[2, 4]$ | $[2, 4]$ |
| | | | Radial SVM | | | |
| $\epsilon_{\text{test}}$ | $[0, 0.286]$ | $[0, 0.286]$ | $[0, 0.214]$ | $[0, 0.286]$ | $[0, 0.286]$ | $[0, 0.214]$ |
| $\text{mean}(\epsilon_{\text{test}})$ | 0.113 | 0.085 | 0.081 | 0.101 | 0.084 | 0.081 |
| $|s_l|$ | $[2, 4]$ | $[2, 4]$ | $[2, 4]$ | $[2, 4]$ | $[2, 4]$ | $[2, 4]$ |
| | | | RF | | | |
| $\epsilon_{\text{test}}$ | $[0.071, 0.286]$ | $[0.143, 0.286]$ | $[0.143, 0.214]$ | $[0, 0.286]$ | $[0.143, 0.214]$ | $[0.071, 0.214]$ |
| $\text{mean}(\epsilon_{\text{test}})$ | 0.216 | 0.188 | 0.182 | 0.175 | 0.203 | 0.146 |
| $|s_l|$ | $[2, 4]$ | $[2, 4]$ | $[2, 4]$ | $[2, 4]$ | $[2, 4]$ | $[2, 4]$ |

Table 5: Sensitivity analysis of meta-parameters $m$ and $\alpha$ for the Leukemia dataset

| | Logistic | | | | | |
|---|---|---|---|---|---|---|
| | $m = 5,000$ | | | $m = 10,000$ | | |
| | $\alpha = 0.1$ | $\alpha = 0.05$ | $\alpha = 0.01$ | $\alpha = 0.1$ | $\alpha = 0.05$ | $\alpha = 0.01$ |
| $\epsilon_{\text{test}}$ | $[0, 0.42857]$ | $[0, 0.28571]$ | $[0, 0.21429]$ | $[0, 0.5]$ | $[0, 0.35714]$ | $[0, 0.21429]$ |
| $\text{mean}(\epsilon_{\text{test}})$ | 0.12519 | 0.09505 | 0.06505 | 0.12548 | 0.10005 | 0.06505 |
| $|s_l|$ | $[2, 4]$ | $[2, 4]$ | $[2, 4]$ | $[2, 4]$ | $[2, 4]$ | $[2, 4]$ |

Table 6: Sensitivity analysis of meta-parameters $m$ and $\alpha$ for the Leukemia dataset with simple logistic and AIC for test error estimation

Table 7: Sensitivity analysis of meta-parameters $m$ and $\alpha$ for the Meter A dataset with Linear-Kernel SVM.

| | $m = 10,000$ | | | $m = 40,000$ | | |
|---|---|---|---|---|---|---|
| | $\alpha = 0.2$ | $\alpha = 0.1$ | $\alpha = 0.05$ | $\alpha = 0.2$ | $\alpha = 0.1$ | $\alpha = 0.05$ |
| $\epsilon_{\text{test}}$ | $[0.056, 0.500]$ | $[0, 0.111]$ | $[0, 0.056]$ | $[0.056, 0.722]$ | $[0, 0.667]$ | $[0, 0.056]$ |
| $|s_l|$ | $[5, 6]$ | $[2, 6]$ | $[2, 6]$ | $[4, 6]$ | $[2, 6]$ | $[2, 6]$ |

Table 8: Colon dataset: Correlation Matrix of Variable Importance

|        | Lasso       | L-SVM       | R-SVM       | RF          |
|--------|-------------|-------------|-------------|-------------|
| Lasso  | 1.00000000  | 0.74568151  | -0.05654834 | -0.05374952 |
| L-SVM  | 0.74568151  | 1.00000000  | -0.03513071 | -0.06136117 |
| R-SVM  | -0.05654834 | -0.03513071 | 1.00000000  | 0.28045757  |
| RF     | -0.05374952 | -0.06136117 | 0.28045757  | 1.00000000  |

Table 9: Leukemia dataset: Correlation Matrix of Variable Importance

|        | Lasso       | L-SVM       | R-SVM      | RF          |
|--------|-------------|-------------|------------|-------------|
| Lasso  | 1.00000000  | 0.89133907  | 0.7865345  | -0.04117896 |
| L-SVM  | 0.89133907  | 1.00000000  | 0.8179987  | -0.01653499 |
| R-SVM  | 0.78653446  | 0.81799867  | 1.0000000  | 0.13001335  |
| RF     | -0.04117896 | -0.01653499 | 0.1300133  | 1.00000000  |

In support of the generally consistent performance across model classes and datasets, we extracted information on the frequency of variable selection (variable importance) across different classes within the same datasets analyzed in this work and looked at the correlation between these frequencies. Tab. 8 and 9 show these analyses for the Colon and Leukemia datasets respectively. For the Colon dataset, we see a high degree of correlation between Lasso and L-SVM which build linear decision boundaries while there is a less strong (but still reasonable) degree of correlation between R-SVM and RF which produce non-linear decision boundaries (whereas the correlations between these two groups of model classes are very low). We notice a similar pattern for the Leukemia dataset where however there is also a high degree of correlation of the R-SVM with the linear boundary models and a lower degree of correlation (but still non-negligible) with the non-linear model RF. What this seems to suggest is that, conditional on different non-linear patterns in the data, the identification of important predictive variables by the SWAG is relatively consistent across model classes (especially if these classes share some underlying assumptions): they select the same variables and consider the same variables to be important. Since these results are based on variable importance, there is a good reason to believe that these results would hold even for different choices of $\alpha$ and $m$ since they do not greatly affect the detection of important variables once they are screened in the first step. In conclusion, returning to the choice of the SWAG meta-parameters, knowing that $p_{\max}$ is chosen by the user for sparsity preferences (and not for performance), $\alpha$ should generally be chosen small enough (depending also on the number of variables $p$) and $m$ can be made as large as computational power allows: then the user can post-process the SWAG models as strictly as they want to pick the best performing ones. As stated earlier, similar reasoning must be applied, for example, also for the choice of the meta-parameters of Rashomon sets, i.e. reference model $l^*$ and parameter $\theta$, where the reference model is usually a pre-existing empirical risk minimizer chosen by the user.

# E  Empirical Results

Here we give further details regarding the empirical results presented in Sec. 3. More specifically, we provide (i) more details regarding the logic used to determine the SWAG meta-parameters, (ii) the training error ($\epsilon_{\text{train}}$) results, number models and Jaccard index description for the considered datasets, and (iii) the details regarding the variables included in the SWAG networks for the Meter A and LSVT data.

## E.1  SWAG Parameter Choice

The choice of the SWAG meta-learning parameters is made in line with the rules of thumb described in Sec. 2.1 and with the intent of reducing overall computational time while ensuring a reasonable exploration of the variable space. More specifically, we start by defining $p_{\max}$ based on the EPV rule discussed in van der Ploeg

et al. (2014) and, for each dataset, we verify that the selected $p_{\max}$ reaches at least an EPV $> 4$ as suggested by Vittinghoff & McCulloch (2007). Based on this, we define $m$ with the aim to at least explore all the subspace of two-dimensional models generated by $p^\star$ (i.e., $\binom{p^\star}{2}$). To determine this value in practice, we first evaluate $\binom{p}{2}$ and, given the computational power at our disposal, we select a given percentage of this quantity that we may call $m_2$ since it refers only to two-dimensional models. This choice then automatically determines a value $p^\star$ that satisfies our original aim since $p^\star = 1+\sqrt{1+8m_2}/2$. The value of $\alpha$ follows automatically and it is set to $\alpha = p^\star/p$ or, if this value is not appropriate for the user, the parameter $m_2$ can be varied in order to obtain the desired value of $\alpha$. In particular, we find that the range $0.01 \le \alpha \le 0.05$ is a good compromise between fixing an $\alpha$ as small as possible (in order to select strong models) and the possibility of exploring a considerable portion of the variable space. Taking this into account, in order to determine the final value of $m$, we just need to consider the growing dimension of the model up to $p_{\max}$. To do so, we multiply the evaluation made at the two-dimensional space of models by $p_{\max}$ and obtain our final $m = m_2 \, p_{\max}$. It implies that, following our choices, $m$ is a linear function of $p_{\max}$.

## E.2 SWAG Training Error

For additional information purposes, we provide results on the training error ($\epsilon_{\mathrm{train}}$) observed for the SWAG models and their original versions which are represented in Fig. 4. As can be observed, the training error ranges of the SWAG models are generally lower than that of the original model versions. We expect this since the SWAG models are selected based on a cross-validation criterion evaluated on the training data (which is not the case for the original versions).

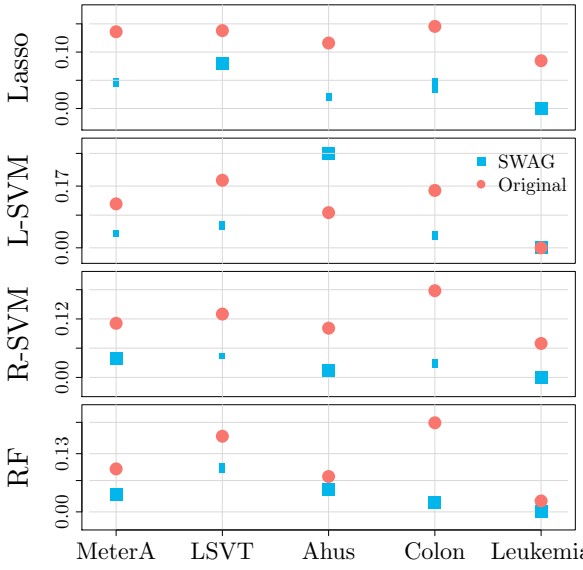

Figure 4: SWAG training error range $\epsilon_{\mathrm{train}}$ (blue rectangles) and original learning method training error (red dots). The blue *squares* imply that the SWAG models all have the same training error.

## E.3 Number of SWAG Models

Tab. 10 collects the information on the number of SWAG models $|\widetilde{\mathcal{M}}|$ per model class and dataset shown in this work (see Tab. 1). As we can see, the number of models varies between datasets and model classes, indicating whether there can exist sparse models as a result of the Rashomon Effect. For example, a number that stands out is the number of models based on the Lasso class, i.e. 5, which can suggest that models belonging to its Rashomon set are of larger dimensions than $p_{\max}$. This is consistent with Fig. 1 where we can see that the original model performs better than the corresponding SWAG models.

Table 10: Number of models in the SWAG libraries

|        | MeterA | LSVT | Ahus   | Colon | Leukemia |
|--------|--------|------|--------|-------|----------|
| Lasso  | 32     | 5    | 720    | 64    | 157      |
| L-SVM  | 74     | 417  | 447    | 66    | 98       |
| R-SVM  | 26     | 28   | 1'137  | 70    | 40       |
| RF     | 26     | 27   | 637    | 59    | 20       |

### E.4  Jaccard Index Computation

We briefly explain how the Jaccard index was computed to present the results in Tab. 2. Generally speaking, we pick two models belonging to the SWAG library and then apply the Jaccard index to the sets of variables included in these two models: we then do so for all pairs of SWAG models. More specifically, for sets $A$ and $B$ the Jaccard index is given by:

$$J(A, B) = \frac{A \cap B}{A \cup B}.$$

Therefore, as an example, if $A = \{2, 3\}$ is the index set of variables in one SWAG model (i.e. this model contains variables 2 and 3) and if $B = \{2, 3, 12\}$ is the index set of variables in another SWAG model, then the Jaccard index is $2/3 \approx 0.67$ denoting a considerable overlap in variables between the two models. This is computed for each pair of models in the SWAG sets therefore generating a collection of Jaccard indexes which we summarize with their median value and range for each dataset and model class.

### E.5  Rashomon Simulation: Latent Structure

To the best of the authors' knowledge, there are no well established strategies for simulating from multi-model settings (e.g. simulating Rashomon sets), aside from adding noise to the data which however reduces possibility of recovering the true model. Therefore, given the motivation for the SWAG, we consider a latent factor setting where few latent variables are the true predictors for a binary response $Y$. However, we assume that we subsequently do not observe the latent variables that generate the response, but only observe some observed (manifest) variables that are linked to the true unobserved latent predictors. Following this structure, it is therefore possible to have different combinations of correlated (manifest) variables that predict the response almost equally well.

More in detail, we generate $n = 2000$ independent observations and $p = 300$ candidate manifest predictors. The underlying signal is driven by $p_{\text{latent}} = 5$ latent variables, each of which (mainly) influences $p_{\text{obs}} = 3$ observed variables, leading to a total of 15 informative predictors (out of the $p = 300$ total ones): this corresponds to the sparse settings that the SWAG generally targets. The latent variables are simulated as

$$X_{\text{latent}} \sim \mathcal{N}\left(0, I_{p_{\text{latent}}}/p_{\text{latent}}\right),$$

where $I$ represents the identity matrix, and the binary outcome follows a logistic model

$$\Pr(Y = 1 \mid X_{\text{latent}}) = \frac{1}{1 + \exp\left[-(1 + X_{\text{latent}}\boldsymbol{\beta})\right]},$$

where $\boldsymbol{\beta} = (\beta_1, \ldots, \beta_{p_{\text{latent}}})^\top$ is a perturbed constant vector, i.e. $\beta_j = 2 + \varepsilon_j$ with small random deviations $\varepsilon_j$ from a uniform distribution to introduce mild heterogeneity across latent effects finally delivering i.e. $\boldsymbol{\beta} = [1.981241, 1.989770, 2.005828, 2.032657, 1.976135]$. The response is then generated as $Y_i \sim \text{Bernoulli}(\Pr(Y_i = 1))$ for $i = 1, \ldots, n$. Each latent variable prevalently gives rise to three correlated observed variables through a structured factor model

$$X_{\text{obs}} = X_{\text{latent}} U^\top + R,$$

where $U$ is a block-diagonal factor-loading matrix of dimension $(p_{\text{latent}} \times p_{\text{latent}} p_{\text{obs}})$. Each block of $U$ corresponds to one latent variable and consists of a vector of weights $(1, 1, 1) + \varepsilon$, where $\epsilon$ is a vector of three

independent random perturbations (again uniform) added to each entry. These perturbations act as small random noise terms that prevent perfect collinearity between observed variables derived from the same latent factor and allow other latent variables to be slightly correlated to all latent factors. More specifically, each latent factor has three manifest variables that are highly correlated to it, but all other manifest variables also have a very low level of correlation to it as well. Although the general correlation structure is still sufficiently well separated between latent factors, we believe that adding these perturbations can make this setting slightly more realistic. The additive noise matrix $R$ introduces additional independent Gaussian noise,

$$R_{ij} \sim \mathcal{N}(0, \sigma^2), \qquad \text{with } \sigma = 0.1,$$

As mentioned above, this construction induces strong within–cluster correlations among variables linked to the same latent dimension, while keeping inter-cluster correlations weak. As a result, several distinct subsets of predictors contain overlapping information about the response, creating a non-unique collection of sparse models with nearly equivalent predictive performance.

Again as mentioned earlier, to emulate realistic high-dimensionality, we augment the design matrix with $p_0 = p - p_{\text{latent}} p_{\text{obs}} = 285$ additional noise features, each drawn from $\mathcal{N}(0, 1/p_0)$. The complete predictor matrix is therefore

$$X = [\, X_{\text{obs}} \mid X_0 \,],$$

where $X_0$ represents the matrix of uninformative predictors. The simulation is replicated $B = 100$ times. For each replicate $b = 1, \ldots, B$, a new dataset $(X^{(b)}, Y^{(b)})$ is generated and the SWAG algorithm is applied under a binomial model with maximum dimension $p_{\max} = 20$ and percentile $\alpha = 0.1$. Each simulation run will generate a library of models, $\{\widehat{\mathcal{M}}_b\}_{b=1}^B$, and we will focus on different metrics with respect to what could be considered a reasonable Rashomon set. Again, the SWAG does not aim to find Rashomon sets, but it is interesting to understand how it behaves if one were to be defined. More specifically, we will consider the reference model $l^*$ to be the fitted logistic regression on the *true unobserved latent variables*: this is the true model that we therefore fit using the true variables. In practice this would not be possible since we would not have access to the true latent factors, however for the purpose of investigation we take this to be the reference model. Based on this, the AIC values for these true fitted models range between 1617 and 1860, with mean and median at 1730 and 1733 respectively, and with lower and upper quartile roughly $\pm 40$ from the median: this suggests a normal distribution for the AIC values of the true fitted model. We take the $\theta$ parameter (which defines the "almost-equivalence" of models) to be the standard deviation of these AIC values, i.e. $\theta \approx 50$. Therefore, for each simulation, we will check if the library of models selected by SWAG is contained in the Rashomon set $\mathcal{R}_\theta(l^*) := \{l \in \mathcal{L} \mid D(l) \leq D(l^*) + \theta\}$ where $l^*$ is the true fitted model in each simulation run. Hence the value of $\theta$ is fixed across simulations, but the reference model changes (i.e. the true fitted model on each run). Notice that we are therefore using the best possible fitted model as a reference each time, which is quite demanding to compete with for other models that only rely on noisy copies of the original latent variables. Applying the same post-processing rule as in the experiments in the main text, **we observe that 100% of the SWAG models are included in the Rashomon set in every run**. This indicates, in this *limited* simulation setting, that the SWAG can recover a subset of models in the Rashomon set.

Having verified this, we consider looking at other metrics of relevance for the SWAG libraries within our latent factor setting. Firstly, since with the SWAG we screen approximately $\alpha \, p$ variables, we have 30 screened variables to run through the general step each time. In all simulations, all 15 manifest variables were always selected along with other 15 irrelevant variables (spuriously correlated to the response). Moreover, after post-processing, all models in the libraries were of dimension $p^* \geq 5$ indicating that the minimum dimension for the SWAG library is the actual dimension of the latent factors. With this in mind, focusing on the 15 manifest variables (i.e. loading on the latent factors) we consider a few metrics that we consider relevant for this purpose: (i) the proportion of models of dimension $p^* = 5$ in the SWAG library that contain manifest variables coming from distinct latent factors; (ii) the number of manifest variables in the top-15 variables in terms of frequency (i.e. number of models in which each variable appears within the set); (iii) the number of manifest variables coming from distinct latent factors in the top-5 variables in terms of frequency. The first metric (Prop. Distinct for $p^* = 5$) aims to assess to what extent the SWAG models are representative of the underlying distinct latent factors, while the second metric (Manifest in Top 15) aims to understand how well the SWAG frequencies detect the importance of the manifest variables. Finally, the third metric (Distinct in

Top 5) is someway similar to the first but measured on the top-5 most frequent variables, where we would hope that the top-5 variables contain manifest variables representing distinct latent factors. Tab. 11 provides summary statistics on these metrics across the 100 runs.

Table 11: Summary Statistics for Latent Simulation. Minimum, first quartile, median, mean, third quartile and maximum (columns) for the three metrics considered (rows) over the 100 simulations.

|  | Min. | 1st Q. | Median | Mean | 3rd Q. | Max. |
|---|---|---|---|---|---|---|
| Prop. Distinct for $p^* = 5$ | 0.1565 | 0.3781 | 0.5162 | 0.5565 | 0.7218 | 1.0000 |
| Manifest in Top 15 | 5.00 | 7.00 | 8.00 | 7.93 | 9.00 | 10.00 |
| Distinct in Top 5 | 3.00 | 4.00 | 4.00 | 4.15 | 5.00 | 5.00 |

Firstly, from the first row of Tab. 11 we can observe that in at least half the runs, the SWAG libraries contained more than 50% of models of dimension 5 that were built using manifest variables from distinct latent factors. Given that models in the following dimensions are built on those of dimension 5, this suggests that a significant component of the SWAG libraries are developed on variable combinations that are representative of the true underlying factors. Following this, from the second row we observe that at least 5 manifest variables are always present in the top-15 most frequent variables across all runs. This is comforting since we would hope that at least five of them are present to represent the true underlying dimension. In addition to this, we see that on average there are 8 manifest variables and a maximum of 10 in the top-15 across all runs: this can also be considered positive since we would not want all 15 manifest variables in the top-15. More specifically, assuming the top-5 contain mostly manifest variables representing distinct latent factors, then we would expect the other manifest variables to be selected randomly (conditioned on the first 5): indeed, having 10 manifest variables left (out of the 25 screened variables left) the probability of selecting a manifest variable out of the remaining 10 positions in the top-15 is 0.4 leading to an expected average of 9 manifest variables among the top-15. We see that the observed average is slightly lower, but this is justified by the results in the third row where we see that in the vast majority of runs, the top-5 most frequent variables represent at least 4 distinct latent factors (with a minimum of 3). Overall, for the restricted setting of this simulation, we can observe how the SWAG appears to find a subset of the Rashomon set and, based on this, is able to well represent the true underlying latent structure for this problem.

### E.6   SWAG Network Details

In this section, we reproduce the same analyses and interpretations included in the main manuscript (for ease of readability) and add the details regarding the variables included in the SWAG networks for the Meter A and LSVT datasets.

### E.6.1   Meter A

The Meter A data is analyzed in Gyamfi et al. (2018) and collects measurements on ultrasonic flowmeter diagnostics. Achieving good diagnostics regarding the health of a flowmeter is of extreme importance for condition-based maintenance in many industrial sectors such as the oil and gas industry since incorrect measurement can entail considerable economic and material losses (see, e.g., TUV-NEL, 2012). In this data, the goal is to classify the health of a meter into two classes: "Healthy" (Class 1) or "Installation effects" (Class 2). Given that the variables are measurements of physical nature, we decide to consider all first-order interactions which finally delivers a total variable size of $p = 666$ (36 original variables plus $\binom{36}{2} = 630$ interactions). Using the results from Lasso-based SWAG, we can see how the variables most frequently included in the selected models can be arranged into a network where the edges represent the most common connections between these variables as can be seen in Fig. 5. Therefore, in order to understand the mechanics and perform diagnostics for this ultrasonic flowmeter, among the 666 variables, a researcher could for example focus their attention on the interaction (i) between flatness ratio and gain as well as (ii) between the speed of sound and gain at the first end (of the fifth path). In particular, a consistent flatness ratio across various paths indicates structural integrity and alignment, crucial for optimal performance. Variations in the flatness ratio can suggest issues such as misalignment or internal obstructions, which can significantly impair the

gain readings and, consequently, the flow measurements. The flatness ratio can affect wave propagation, with deviations leading to distorted signals. This distortion, which manifests in reduced gain measurements, indicates latent defects that could progress into more severe operational failures if not addressed (Tang et al., 2015). In addition, changes in the speed of sound can indicate alterations in fluid properties—such as temperature, pressure, or contamination levels (which may not be immediately observable). Monitoring the speed of sound alongside gain measurements can uncover underlying problems within the fluid system that affect measurement accuracy (Bombarda et al., 2021). As a final note, given that cost-based maintenance can have asymmetric costs according to the decision taken on the flowmeter, the SWAG could allow to select models based on the corresponding (non-convex) cost-function instead of a symmetric kind of loss (i.e., each type of error is weighted equally) that model classes are usually trained on (see, e.g., Crone, 2002; Masnadi-Shirazi & Vasconcelos, 2007).

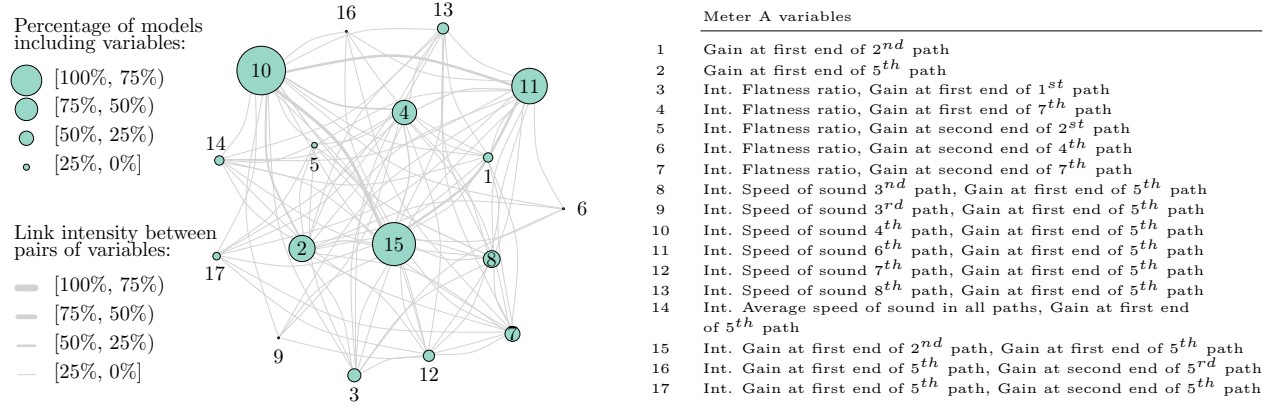

Figure 5: Network of variable importance and pairwise link-intensity in the set of models $\widetilde{\mathcal{M}}$ for the Meter A dataset using the Lasso-based SWAG results. "Int." denotes the interaction between two variables

| Variable | Frequency | Degree Proportion | Betweenness | Eigen Centrality |
|---|---|---|---|---|
| 10 | 32 | 0.2010 | 8.3084 | 1.0000 |
| 15 | 29 | 0.1801 | 6.6084 | 0.9731 |
| 11 | 20 | 0.1101 | 11.9417 | 0.4764 |
| 2 | 14 | 0.0909 | 55.2696 | 0.3822 |
| 4 | 13 | 0.0822 | 4.9917 | 0.3405 |
| 8 | 11 | 0.0769 | 14.5333 | 0.2848 |
| 7 | 6 | 0.0420 | 17.7552 | 0.0614 |
| 3 | 4 | 0.0280 | 0.0000 | 0.0402 |
| 13 | 4 | 0.0280 | 2.3000 | 0.0405 |
| 12 | 4 | 0.0280 | 0.0000 | 0.0402 |
| 1 | 3 | 0.0210 | 3.6357 | 0.0178 |
| 14 | 3 | 0.0210 | 1.5917 | 0.0149 |
| 5 | 2 | 0.0140 | 19.8738 | 0.0067 |
| 17 | 2 | 0.0140 | 1.0762 | 0.0108 |
| 16 | 2 | 0.0140 | 8.6004 | 0.0101 |
| 6 | 2 | 0.0140 | 1.7409 | 0.0109 |
| 9 | 1 | 0.0070 | 3.1000 | 0.0026 |

Table 12: Network statistics for the SWAG library of Meter A dataset. These statistics include variable frequency, degree proportions, betweenness, and eigen centrality.

Table 12 provides insights into the properties of the network through some basic statistics. More specifically, in this network variables 10 and 15 appear to be the most central nodes: both have the highest frequencies

and eigen centralities, suggesting that they are highly stable and jointly form the structural basis of the predictor space. They consistently co-occur with a wide range of other variables, indicating that they are compatible across different models and likely play a fundamental role in predictive performance. Variable 11, while less frequent, maintains a strong eigen centrality, placing it in close proximity to the core and implying that it frequently co-occurs with these dominant predictors, reinforcing their joint explanatory role. By contrast, variable 2 displays an exceptionally high betweenness score despite a moderate frequency, implying that it serves as a bridge between clusters of predictors that are otherwise rarely connected. This bridging function may indicate that variable 2 captures a general effect or contextual factor that connects distinct modeling scenarios. Variable 7 also exhibits elevated betweenness relative to its modest frequency, reinforcing its role as a connector across clusters, though less globally influential than variable 2. Conversely, variables such as 4 and 8 have moderate centralities and appear to participate in specific, well-defined clusters: they contribute locally without exerting a strong effect on the overall network. A particular case is variable 5, which shows high betweenness centrality despite low frequency and degree. Indeed, although variable 5 appears in only a couple models, these models tend to combine predictors from distinct clusters, for instance, the core group of highly stable variables (10 and 15) with more specialized subsets (such as variables 4, 7, or 8). As a result, variable 5 lies on many of the shortest paths between clusters, yielding high betweenness values. In practical terms, such a variable may not be universally important but allows different modeling regimes to share information. Its presence highlights how marginally unimportant predictors can still play disproportionate structural roles in maintaining the connectivity and interpretability of the overall SWAG library.

At the periphery of this network, variables 1, 6, 9, 14, 16 and 17 show low values across all centrality metrics. Their rare inclusion and weak connections suggest that they play a specialized or redundant role, potentially associated with specific modeling contexts but not essential to the general predictive structure. The combination of high-frequency nodes (10, 15), bridging intermediaries (2, 7, 8), and peripheral specialists (e.g., 1, 5, 16) suggests a multi-layered organization of the SWAG library: a tightly connected core of informative predictors, surrounded by a set of variables that either connect different modeling approaches or contain context-specific information.

### E.6.2 LSVT

The voice rehabilitation dataset is analyzed in Tsanas et al. (2013) in order to assess the effectiveness of a computer program called "Lee Silverman voice treatment (LSVT) Companion" which allows patients with Parkinson's disease to independently progress through a rehabilitative treatment session. Taking data on 126 samples from 14 patients who followed the latter treatment, 310 dysphonia measures are taken on each of them (plus information on sex and age of the patients) and used to understand if they could correctly predict whether the patients' voices are "acceptable" or "unacceptable" after this treatment. In their analysis, a robust feature selection is used to select 8 variables (based on the first eight variables classified by the feature selection method) and subsequently R-SVM is tested (along with RF) to obtain around 90% accuracy in classifying patients' progress.

There is also scientific interest in determining the variables (and combinations thereof) that most contribute to the definition, in this case, of a Parkinson's disease (PD) speech treatment as being acceptable or not. Also in this case, the SWAG models (based on the R-SVM) can be arranged into a network in order to allow for interpretation as seen in Fig. 6. Based on the SWAG network, researchers interested in improving speech treatment should focus on the $2^{nd}$ and $3^{rd}$ Mel-Frequency Cepstral coefficients (MFCC) and on the entropy with base-4 logarithmic coefficients (as well as the interactions between these three variables as highlighted by their frequent presence in the same SWAG models). In particular, MFCCs such as the $2^{nd}$ and $3^{rd}$ coefficients, are pivotal in characterizing the spectral properties of sound produced by the vocal tract. They provide details about vowel sounds and overall speech quality, making them useful for assessing the impact of LSVT therapy. Moreovwer, various jitter measurements (such as Jitter F0 absolute perturbation and Jitter F0 range across the 5th and 95th percentiles), are crucial in assessing vocal stability: with heightened jitter often correlating with less stable vocal production where this instability can signal underlying physiological issues affecting vocal fold function, suggesting a need for targeted therapeutic interventions. For instance, a patient exhibiting low jitter values and consistent MFCC readings alongside stable entropy measures is likely

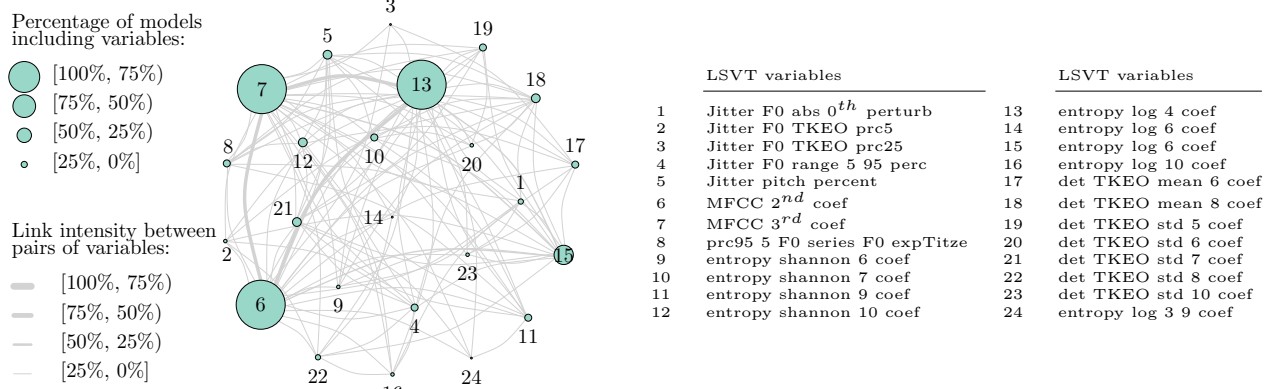

Figure 6: Network of variable importance and pairwise link-intensity in the set of models $\widetilde{\mathcal{M}}$ for the LSVT dataset using the Radial-Kernel SVM SWAG results.

progressing well in their treatment. Conversely, high jitter or low entropy may indicate challenges that require further intervention. These attributes serve not only as indicators of progress but as quantifiable measures for tailoring personalized therapy sessions, enhancing the overall management of voice characteristics affected by PD. Each attribute interrelates, influencing the others: for example, stabilization in jitter can lead to improved MFCC values as patients develop stronger vocal control through therapy.

| Variable | Frequency | Proportion | Betweenness | Eigen Centrality |
|---|---|---|---|---|
| 13 | 28 | 0.2024 | 58.3863 | 1.0000 |
| 7 | 28 | 0.2024 | 58.3863 | 1.0000 |
| 6 | 28 | 0.2024 | 58.3863 | 1.0000 |
| 15 | 6 | 0.0377 | 18.9850 | 0.0554 |
| 21 | 5 | 0.0317 | 15.0353 | 0.0354 |
| 5 | 4 | 0.0238 | 14.6050 | 0.0200 |
| 18 | 4 | 0.0238 | 8.7256 | 0.0200 |
| 12 | 3 | 0.0238 | 11.9732 | 0.0200 |
| 19 | 3 | 0.0238 | 26.5159 | 0.0200 |
| 10 | 3 | 0.0238 | 8.0701 | 0.0200 |
| 17 | 2 | 0.0159 | 5.5554 | 0.0089 |
| 11 | 2 | 0.0159 | 21.9490 | 0.0089 |
| 4 | 2 | 0.0159 | 9.9414 | 0.0089 |
| 8 | 2 | 0.0159 | 5.5554 | 0.0089 |
| 1 | 2 | 0.0159 | 5.6700 | 0.0089 |
| 22 | 2 | 0.0159 | 6.2286 | 0.0089 |
| 9 | 2 | 0.0159 | 9.1065 | 0.0089 |
| 2 | 2 | 0.0159 | 5.5554 | 0.0089 |
| 24 | 2 | 0.0139 | 0.0000 | 0.0089 |
| 16 | 1 | 0.0079 | 9.8200 | 0.0022 |
| 3 | 1 | 0.0079 | 7.3840 | 0.0022 |
| 20 | 1 | 0.0079 | 7.6672 | 0.0022 |
| 14 | 1 | 0.0079 | 18.7112 | 0.0022 |
| 23 | 1 | 0.0079 | 8.7083 | 0.0022 |

Table 13: Network statistics for the SWAG library of LSVT dataset. These statistics include variable frequency, degree proportions, betweenness, and eigen centrality.

More specifically, looking at Tab. 13, we can observed an evident core-periphery structure. Variables 13, 7, and 6 dominate the network: each exhibits the highest frequency, degree proportion, betweenness and eigen centrality (all equal to 1.0). These three variables therefore constitute the *core set* of the network. Their consistent inclusion across nearly all models and their dense interconnection suggest that they form a stable predictive structure (a combination of variables that co-occur everywhere and are the basis for the model library). Their high eigen centralities further indicate that they are not only central by count but also by influence, being directly connected to other highly central variables. Beyond these variables, variables 15, 21, and 5 show moderately high frequencies (6, 5, and 4 respectively) and intermediate betweenness values (roughly 15–19), qualifying them as *secondary connectors* between the core and the mid-peripheral region of the network. These variables likely represent complementary predictors that frequently accompany the core set but also extend to other model subsets, thereby enhancing model diversity. Variable 19 stands out with a notable betweenness value (26.52) despite moderate frequency, implying a bridging function: it appears in models that link otherwise distinct subsets of predictors, potentially mediating between the core (13–7–6) and more specialized configurations. Variable 12 also plays an intermediate role, having comparable betweenness (11.97) and moderate eigen centrality, which suggests that it supports connectivity among local clusters surrounding the core group. The remainingvariables (including 10, 18, 17, 11, 4, 8, 1, 22, 9, and 2) exhibit similar moderate proportions and low to moderate eigen centralities (around 0.009). Their pattern of co-occurrence indicates localized interactions: they tend to form small, context-specific models that are built around the central three variables. Although they contribute little to the overall structure of the network, they likely represent domain-specific or situational predictors that enrich the model library under certain parameterizations. Finally, the peripheral variables (i.e. 16, 3, 20, 14, and 23) appear rarely and have minimal degree, betweenness, and eigen centrality. Their sparse inclusion and weak connectivity suggest that they function as highly specialized or redundant features. Despite their low centrality, their occasional high betweenness values (e.g., variable 14 with 18.71) imply that even peripheral variables can temporarily act as bridges when present, connecting otherwise unlinked model subgroups. Overall, the LSVT network demonstrates a clear hierarchy: a densely connected core (13–7–6) of highly stable variables, an intermediate layer of partially integrative predictors (e.g., 19, 15, 21), and a peripheral set of specialized variables.

