# OpenReview forum: "More of Less: A Rashomon Algorithm for Sparse Model Sets"
_TMLR — Rejected by TMLR_

### Review · Reviewer_HnAY · 2025-09-25

**Summary Of Contributions:**

This paper introduces the Sparse Wrapper AlGorithm (SWAG), a novel heuristic method aimed at identifying multiple, sparse, yet well-performing models from high-dimensional data, in the context of the Rashomon effect. The algorithm is model-agnostic and uses a greedy forward-stepwise approach to build candidate models with an increasing number of features. The authors claim the main contributions are providing users with a diverse set of interpretable, sparse models with predictive power comparable to more complex models, offering flexibility in model selection, and enabling insights into variable importance and interactions through network analysis of the resulting model set. They support these claims with empirical evaluations across several datasets and different base model types, including Lasso, SVMs, and Random Forests.

**Audience:**

Yes

**Audience Explanation:**

Yes, the findings of this paper would likely interest a significant portion of the TMLR audience. The work sits at the intersection of several key areas in machine learning, including model interpretability, model selection, and understanding model multiplicity (the Rashomon effect). Given the growing emphasis on trustworthy and understandable AI, methods that can produce simpler yet accurate models are highly relevant. Researchers working on feature selection, model-agnostic techniques, and the stability and robustness of machine learning predictions would find the proposed SWAG algorithm and its analysis of sparse model sets quite pertinent. The approach offers a practical tool for exploring trade-offs between sparsity, performance, and model diversity, which is a common challenge in many applied ML domains.

**Broader Impact Concerns:**

The work primarily presents a methodological advancement in finding sparse model sets and does not appear to have direct ethical implications requiring a specific Broader Impact Statement. The ethical considerations are largely general to any model selection technique, such as the potential for models to reflect biases present in the training data or the risks of misinterpretation in sensitive domains. Since SWAG is model-agnostic and data-driven, the responsibility for ethical application, including fairness assessments and careful deployment in high-stakes scenarios, would lie with the user and the specific application context, rather than being inherent to the SWAG algorithm itself.

**Claims And Evidence:**

Yes

**Claims Explanation:**

The claims made in this submission regarding the SWAG algorithm are generally well-supported by accurate and clear empirical evidence. The authors convincingly demonstrate through experiments on several datasets and across four different model types (Lasso, L-SVM, R-SVM, RF) that SWAG can identify multiple, diverse, and sparse models with predictive performance comparable to, or even better than, models using all features. The model-agnostic design is evident, and the benefits of flexibility and potential for enhanced interpretability through network analysis are plausibly presented. While the method is heuristic and can be computationally intensive, the provided results, including performance metrics, diversity indices, and sensitivity analyses, offer solid backing for the effectiveness of SWAG in generating useful sets of sparse models.

**Requested Changes:**

Here are a few but there could be more:

1.  **Clarification on Baseline for Rashomon Set Definition (Strengthening):** While SWAG doesn't strictly adhere to the typical Rashomon set definition requiring a reference model `l*` and a threshold `θ` (as discussed in Sec 2.2), it would be beneficial to include a more direct comparison. For instance, the authors could select a high-performing model from their SWAG set (or the full model) as `l*` and empirically show what `θ` range their SWAG models fall into. This would better connect SWAG to the existing Rashomon literature and quantify the "almost-equal performance."

2.  **Deeper Dive into Network Insights (Strengthening):** The network visualizations are a nice contribution. To make the insights more concrete, the authors could:
    *   Quantify network properties (e.g., centrality measures for variables, edge weight distributions).
    *   Provide more detailed examples of how these network structures translate to actionable domain knowledge, perhaps contrasting interpretations from different models within the SWAG set for the same dataset.
    *   Briefly discuss how these network-based insights differ from standard variable importance measures.

3.  **Computational Cost Analysis (Strengthening):** The paper acknowledges SWAG is computationally intensive. It would be valuable to provide more concrete runtime comparisons in the main text or a more prominent appendix section, perhaps against some of the baseline methods from Table 3, not just different SWAG configurations. While Appendix B.1 has some timings, showing the cost increase relative to single model selection methods would be informative for potential users.

4.  **Comparison with other Set-Producing Methods (Strengthening):** While Appendix A compares SWAG to single-model selection, a discussion or empirical comparison with other methods that *do* produce sets of models (even if not sparsity-focused) could further highlight SWAG's unique contributions. Examples might include different ensemble subset selection techniques or methods focused on model diversity.

5.  **Heuristic Limitations Discussion (Strengthening):** The authors mention the greedy nature. Expanding slightly on the potential limitations would be good – e.g., are there scenarios or data structures where the greedy approach is likely to miss important sparse model sets? This manages expectations and suggests future work.

6.  **Code Availability (Strengthening):** While not explicitly required for the review, stating intent to release code would strengthen the paper's impact and reproducibility, aligning with TMLR's ethos.

---

> ### Author Response · Authors · 2025-11-16
>
> We thank the reviewer for the detailed and constructive feedback. We greatly appreciate the positive assessment of the paper as well as the targeted suggestions for strengthening it. We address the points in order:
>
> 1. **Clarifying the Rashomon baseline.**
>    We agree this is an important way to contextualize SWAG within the Rashomon framework. As noted, SWAG addresses the *Rashomon Effect* but does not attempt to *construct* a Rashomon set, since no reference model $l^*$ or predefined $\theta$ is assumed. Nevertheless, the final SWAG library can overlap conceptually with sparse Rashomon sets. In this sense, the reviewer’s suggestion aligns with what we illustrate in Fig. 1: using the full model as a reference, the distribution of SWAG test errors effectively traces the range of $\theta$ values implied by the choice of $\alpha$ (and the post-processing parameter $\delta$). The upper bound of the blue rectangles corresponds to this empirical $\theta$.
>
>    To partially address this comment—and also in response to another reviewer—we added a simulation based on a logistic latent-factor model (Appendix E.5). In this setup, different manifest-variable combinations are similarly predictive, inducing a controlled Rashomon Effect. Using the latent model as the reference, we define $\theta$ a priori and confirm that all SWAG models fall within this Rashomon set, while also recovering latent structure. We are happy to further refine this point based on additional discussion.
>
> 2. **Deepening network-based insights.**
>    We appreciate this suggestion, which aligns closely with the motivation of SWAG. We expanded Sec. 3 and the appendices with additional discussion and tables for the LSVT and Meter A datasets. Beyond node frequency, we now report degree, betweenness, and eigen-centrality, illustrating how variables matter not only through frequency but also through their structural roles—e.g., serving as bridges between otherwise weakly connected model regions, or forming alternative pathways to predictive performance. These insights highlight latent structure and model multiplicity in ways that differ from standard variable-importance measures, which focus on marginal effects. We thank the reviewer for encouraging us to articulate these strengths more clearly.
>
> 3. **Computational cost analysis.**
>    Following the reviewer’s suggestion—and anticipating the next point—in Appendix B.1 we added a comparison with the single-model methods and the Random Subspace Method (RSM), which is the closest heuristic set-producing alternative. RSM is far more computationally efficient (evaluating only 500 models and avoiding 10-fold CV) but, as shown in the revised Appendix A, this comes at the cost of much higher performance variability and less informative model sets for addressing the Rashomon Effect. These results clarify the computational trade-offs.
>
> 4. **Comparison with other set-producing methods.**
>    We agree this comparison is valuable. As noted above, we added RSM as a direct baseline. Using $p_{\max}$ for model dimension and guidelines from Tian & Feng (2021), we observe that although the *best* RSM models occasionally approach SWAG’s lower test-error range, their median and upper errors are substantially higher across datasets (with a slight exception for Colon). Moreover, all RSM models share the same dimension, whereas SWAG yields sparse models of varying sizes—a key practical and interpretational advantage. These findings reinforce that RSM is not tailored to the Rashomon perspective, whereas SWAG consistently identifies high-performing alternatives.
>
> 5. **Heuristic limitations.**
>    We agree on the importance of explicitly acknowledging these. In Sec. 2 of the revised manuscript we now discuss limitations arising from SWAG’s greedy nature, including the possibility of missing relevant variable combinations in early stages, the consequences of approximating an exhaustive search, and the role of post-processing, computational cost, and stability considerations.
>
> 6. **Code availability.**
>    We agree that this would strengthen reproducibility and the code is indeed already available in an open-source package/library which we intend to release publicly. Due to double-blind constraints we have not shared it yet, but it will be made available upon acceptance.
>
> We thank the reviewer again for the thoughtful feedback, which has helped us substantially improve the clarity and positioning of the manuscript.

---

### Review · Reviewer_azbv · 2025-10-15

**Summary Of Contributions:**

Considering the problem of the Rashomon Effect in which multiple models can provide near identical predictions the authors propose the SWAG algorithm to efficiently extract such Rashomon sets of models for which predictions are within a bound \theta in performance difference. The procedure is very similar to forward selection but performing more advanced model search using sequential feature combinations in a forward manner. The approach is divided into three algorithmic framework. A first screening considering one-dimensional models that that predicts sufficiently well to combine features among these identified features (Algorithm 1). A screening algorithm considering all possible models with a subset of variables. The proposed SWAG (algorithm 3) is then a combination of Algorithm 1 grown and refined by Algorithm 2 to produce models with sufficient performance wrt. optimally identified models. The approach relies on three parameters – maximal number of features used p_max, size of variable space exhaustively evaluated m, and quantile of models preserved \alpha. The approach is evaluated considering standard ML models on a series of benchmark datasets considering models trained using all features and extracted subsets (Fig. 1) as well as a evaluated in terms of a proposed Jaccard metric of feature overlap (Table 2) and feature frequency and co-occurrence network (Fig. 2 and Fig 3). The supplementary includes additional explanations and experimentation also comparing the approach to standard feature selection procedures (Table 3) .

**Additional Comments:**

I find the novelty and technical contribution of this work very limited, the experimentation to be weak, and the approach poorly related to existing works.

**Audience:**

Yes

**Audience Explanation:**

The development of approaches for the extraction of Rashomon related sets is interesting and relevant both in terms of robust predictions, uncertainty quantification and explainability as discussed in the manuscript moving beyond single models to ensembles. As such the manuscript well motivates the approach and its relevance to the community, where Rashomon sets have recently attracted research efforts. The manuscript however lacks relating and positioning the presented work to these recent works considering Rashomon Sets including:
https://arxiv.org/abs/2209.08040
https://arxiv.org/abs/2406.03059
https://www.jmlr.org/papers/v24/23-0149.html

**Broader Impact Concerns:**

The nature of the work is foundational in terms of Rashomon sets and there are no immediate broader impact concerns needing to be discussed.

**Claims And Evidence:**

No

**Claims Explanation:**

Strengths:
* The idea of Rashomon sets are interesting and potentially underexplored.
* The approach taken in terms of Algorithm 3 based on algorithm 1 and 2 is sound.
* The methodology includes also new types of evaluation metrics including Jaccard and networks of variable co-occurrences that are simple and a good way of visualizing results of the learned model sets.

Weaknesses:
* The methodology proposed is straightforward and with very limited novelty.
* No theoretical results are derived in terms of guarantees and optimality in terms of Rashomon sets only a description of how the methodology relates to the Rashomon sets informally.
* The experimentation is not well executed and the evaluated results reported without clear comparisons to related approaches, ground truth and alternatives reported in the literature (see below).

My main concern is the novelty and technical contribution of this manuscript which I find very limited. Algorithm 1 is essentially a standard forward step and algorithm 2 a simple screening procedure akin existing screening approaches as also discussed in appendix A. The key difference is essentially to grow a large set of models that predict well as opposed to identifying the single best predictive model in traditional sequential feature search procedures.

The datasets considered does not include ground truth in terms of the optimal set and what are the relevant Rashomon sets. Synthetic studies on simulated data with ground truth Rashomon sets would strengthen the analysis and ground the approach. It would here also be relevant to compare to alternative procedures. I am not aware of this literature but you could compare to the random subspace method of (RSM) Ho cited.

It also seems there are many alternative approaches tackling the same problem that should be compared against, see also:
https://arxiv.org/abs/2209.08040
and comparison methods in this paper (published NeurIPS 2022).

https://arxiv.org/abs/2406.03059
considering the ground truth Rashomon set and closeness to this (Published KDD’24).

and approaches to systematically evaluate feature attribution in the Rashomon set as opposed to the presented evaluation by networks of feature frequency (node size) and co-occurrences (link strengths):
https://www.jmlr.org/papers/v24/23-0149.html

The present manuscript needs to consider all the above prior works and also the experimentations used there to quantitatively compare and evaluate the proposed methods. The present experimentation is very limited and it is unclear how the presented SWAG compares to current SOTA including the above related works.

**Requested Changes:**

The SWAG approach should be compared to existing methodologies including their evaluation criteria and experimental evaluations, see:
https://arxiv.org/abs/2209.08040 (Compare SWAG to the alternative methodologies used as baseline and presented in the work considering their evaluation setup)
https://arxiv.org/abs/2406.03059 (Evaluate in terms of ground truth Rashomon set as suggested in this paper)
https://www.jmlr.org/papers/v24/23-0149.html (relate and compare presented simple feature summary in terms of frequency and co-occurrence networks to the wider effort of feature attribution as well as this recent work on feature attribution in the Rashomon set).

How are the specific datasets used chosen for the experimentation out of a vast number of datasets from UCI machine learning repository and other sources? The dataset selection criteria should be further clarified.

In table 3 of the appendix I understand that 10 fold CV is used to report results, but SWAG also uses test-error to choose models whereas the compared approaches as far as I understand are based on AIC and other complexity penalizations. For fair comparison all methods should use test-error estimations for their selections. Please clarify this. Test-error metrics are widely used for sequential feature selection procedures and standard in many wrappers. It is therefore unclear why a direct comparison to same selection criteria is not considered in this experimentation.

For variable selection L1 regularization paths have been considered. It is unclear why the results in Table 3 in the supplementary are not also considered entire regularization paths of lasso regularization and selecting the variable set with best test-performance which is a standard procedure and more modern approach to feature selection than sequential feature selection. Please include this comparison.

---

> ### Author Response · Authors · 2025-11-16
>
> We thank the reviewer for the detailed and thoughtful review. We address the main points below.
>
> 1. **Clarifying the goal of SWAG and its novelty.**
>    We apologize for not communicating clearly that SWAG does *not* aim to estimate Rashomon sets. Instead, it addresses the *Rashomon Effect* in settings where no reference model $l^*$, no range parameter $\theta$, and no restrictive model-class assumptions are available—particularly in high-dimensional problems where sparsity is expected. We have clarified this conceptual distinction in the revised manuscript.
>
>    Regarding the description of Algorithm 2, we want to ensure there was no misunderstanding: SWAG does *not* evaluate all subsets of a given size. Rather, Algorithm 2 screens only (m) combinations **built from the best models of the previous step**. This dependence across steps is essential, as the previously discovered good models guide the forward exploration.
>
>    Concerning novelty: while we understand the reviewer's perspective, considering that the SWAG uses established ideas from forward search and screening, we also believe that there is no existing method that addresses the Rashomon Effect across *arbitrary model classes* (i.e., wrapper methods) and yields *sparse and accurate* alternative models that enable variable-importance summaries such as frequency, co-occurrence networks, and graph-theoretic centrality. Therefore, although these contributions may not be perceived as novel, we believe they nevertheless provide new flexible tools to address the Rashomon Effect as well as new ways of interpreting variable importance within this paradigm. We also acknowledge the limited technical contribution, but since (among others) the SWAG does not minimize a global criterion, deriving theory for heuristic search methods typically requires (unrealistically) strong assumptions; in contrast, Rashomon sets are well-defined at the population-level around an optimization principle and therefore can be studied theoretically.
>
> 2. **On comparisons to Rashomon sets and synthetic experiments.**
>    Since SWAG does not target Rashomon sets, while informative, we believe that these direct comparisons would not strictly be meaningful (or may erroneously support the idea that the SWAG targets Rashomon sets). Still, when using the full model class as a reference, our experiments (Fig. 1) show that SWAG often returns *sparser models with lower test error* than the reference model, which implies that these models lie within the Rashomon set for any reasonable $\theta$.
>
>    The reviewer suggests simulations with ground-truth Rashomon sets. To our knowledge, no procedure exists to simulate data with multiple true variable combinations that jointly form a Rashomon set. However, because SWAG was designed to reveal *latent sparse structure*, we added a new simulation study (Appendix E.5) based on a logistic latent-factor model. Different combinations of manifest variables produce comparable predictive performance, inducing a controlled Rashomon Effect. Using the latent model as the reference, we define an a-priori $\theta$ and show that **all SWAG models fall within this Rashomon set** while also recovering meaningful latent structure.
>
>    We expanded discussion on variable-importance interpretation through SWAG networks. Beyond frequency, we now highlight degree, betweenness, and eigen-centrality, which reveal how variables co-occur, bridge disconnected model regions, or substitute for one another. These insights differ from standard attribution methods, which focus on marginal variable effects; SWAG highlights *context-dependent* importance via model co-occurrence patterns.
>
> 3. **Dataset selection.**
>    We have clarified this point: datasets were chosen for their high-dimensional nature ($n \ll p$), which matches SWAG’s intended sparse-representation setting.
>
> 4. **Clarifying the use of 10-fold CV.**
>    We revised the manuscript to make clear that 10-fold CV was used *only* for hyperparameter tuning of all compared methods. All performance metrics in Table 3 are test-set (missclassification) errors computed under a common evaluation setup.
>
> 5. **On L1-regularization comparisons.**
>    We interpreted the reviewer as requesting Lasso path–based comparisons. These results already appear in Fig. 1 of the main manuscript, where Lasso’s test errors are shown alongside SWAG’s. If the reviewer intended a different comparison (e.g., selecting the best test-error model along the path), we are happy to add it.
>
> We appreciate the reviewer’s recommendations for additional related work and have expanded the positioning of SWAG within the Rashomon-set literature accordingly. We believe these modifications have improved the clarity and message of the manuscript, so we thank the reviewer for this.

---

> > ### Author Response · Authors · 2025-12-12
> >
> > For the purpose of exploring the reviewer's comment on comparing the SWAG with Rashomon set procedures, we compared the Rashomon sets found by TreeFARMS with the models in the SWAG library of decision trees. However, this comparison is not exactly fair since TreeFARMS (as other Rashomon set procedures) is not directly calibrated for high-dimensional problems, while SWAG is not calibrated for low-dimensional problems (since it would reduce to an exhaustive search) and is not directly built to target Rashomon sets (as formally defined). This being said, we compare these procedures on the LSVT data and, given the high-dimensionality, for TreeFARMS we choose a larger regularization parameter, i.e. 0.1, as well as a rashomon bound multiplier of 0.05. This yields a Rashomon set of 9 decision trees of depth up to 3 (at most three variables). With the same meta-parameters as the original analysis on the LSVT data, we ran the SWAG with decision trees between dimensions 1 to 3 yielding 324 decision trees with errors smaller or equal to those in the Rashomon set of TreeFARMS. More specifically, all 9 models in the Rashomon set are present within the SWAG library of 324 models with similar loss to those in the Rashomon set. While this result cannot be interpreted in a conclusive manner (since the SWAG does not directly target Rashomon sets and TreeFARMS is not specifically calibrated for these data settings), it provides additional suggestions that the SWAG is able to recover many models which happen to be in the Rashomon sets.

---

### Review · Reviewer_wtS6 · 2025-11-07

**Summary Of Contributions:**

The authors propose a method for finding a *set* of sparse models that are good at prediction, rather than a single best model, to allow downstreams users to pick the model best suited for their use case. (For example, some variables may be costly to retrieve at inference time than others, so a user may want to pick a sparse model which does not use that variable even if that variable is highly predictive.)

The proposed method, "Sparse Wrapper AlGorithm" (SWAG) is intended to be agnostic to how each model is learned. Assume there are $p$ variables of interest, and for any subset of variables, we can fit a model from those variables.

The first step is an initial screening. We first build $p$ one-D models that each only use a single variable. Let $\alpha$ be a hyperparam for the desired performance percentile. From these $p$ models, we filter to only the top $\alpha$ models (sorted by error), to form an intermediate model set $\tilde{\mathcal{M}}^\*$, as well as a smaller subset $\mathcal{S}^\*$ of more predictive variables. (Let variables be denoted as $s$)

The general step takes a set of n-dimensional models and builds (n+1)-dimensional ones. Sample a model from our set so far, then fit a new model using the variables of the first model, plus one new variable from our subset $\mathcal{S}^\*$ of more predictive variables. Do this until either every (n+1)-dim model is created, or we hit some pre-defined threshold $m$ for maximum number of models for efficiency. These (n+1)-dim models are filtered again according to a top $\alpha$ percentile. Repeat this process until we hit the maximum dimensionality $p_{max}$, another hyperparam.

Last, we may optionally postprocess the model sets. By now we have $p_{max}$ model sets, one for each dimension, each filtered to the top $\alpha$ percentile within their dimension. We can further filter the models by picking a global error threshold $\epsilon$, filtering each per-dimension set to only be the models under that global error. The paper proposes setting $\epsilon$ by first picking the dimension with lowest median error, then picking a quantile $\delta \in (0,1)$ of performance within that dimension.

This overall can be an expensive procedure, due to how many models need to be trained, and so is primarily tested on using fast-to-fit methods like L-SVM, R-SVM, logistic regression, random forests, etc. On various datasets, the authors study how well SWAG generates models of similar error to the original reference model that L-SVM, R-SVM, etc. would naively generate.

**Audience:**

Yes

**Audience Explanation:**

Paper's incremental approach to building Rashomon sets could be of interest, even if it is more heuristics based than other approaches.

**Claims And Evidence:**

Yes

**Claims Explanation:**

The paper's approach to building Rashomon sets differs from some prior methods which start from a reference model and perturb from there. Instead, SWAG builds a model set from scratch (from 1-D models building up), which works as long as you assume sparse structure exists in the data.

However, it does seem exceptionally expensive to do this kind of search. Although $\alpha$ and $m$ are ways to avoid making the search space blow up exponentially, it leads to a classic explore-exploit tradeoff. I imagine SWAG would perform least well when inter-variable effects are stronger, due to its process of adding 1 variable at a time.

Still, I think this approach to model class building could be of interest to those interested in studying Rashomon sets from a different angle.

**Requested Changes:**

N/A

---

> ### Author Response · Authors · 2025-11-16
>
> We thank the reviewer for their thoughtful feedback and appreciate the positive assessment of our contribution. We respond briefly to clarify a few points, also informed by the comments of the other reviewers.
>
> First, we emphasize that SWAG does **not** aim to estimate a Rashomon set in its formal sense. Instead, it is designed to address the **Rashomon Effect**—the empirical multiplicity of well-performing models—particularly in high-dimensional settings where sparse latent structure is plausible. We have clarified this distinction throughout the revised manuscript.
>
> That said, both the original and newly added results (including a new simulation in Appendix E.5) show that SWAG often identifies **sparse models with test error equivalent to, or even better than, the reference models** for the considered datasets. Although SWAG does not target a Rashomon set, these findings empirically demonstrate that it can recover models lying inside the corresponding sparse Rashomon region, whenever such a region exists.
>
> Second, regarding inter-variable effects and the reviewer’s comment on explore–exploit tradeoffs: we fully agree that SWAG’s greedy construction may miss certain interactions, especially when interactions are strong. To address this, we added explicit discussion of these heuristic limitations in Section 2, in addition to our earlier remarks on computational complexity and stability. Nonetheless, both in the high-dimensional datasets and in the new collinearity-based simulation, SWAG continues to perform well even when interactions or correlated structures are present. This suggests a degree of robustness, although we acknowledge that more complex or non-sparse settings may challenge SWAG’s sequential construction.
>
> We again thank the reviewer for their encouraging evaluation and for highlighting aspects that helped us strengthen the clarity and framing of the revised manuscript.

---

### Decision · Action_Editor_JVEx · 2025-12-15

**Recommendation:** Reject

**Additional Comments:**

See comments above.

**Audience:**

No

**Audience Explanation:**

While the reviewers have agreed that the paper may possibly be of interest to some readers, it remains unclear to which readers. Which would be the target audience? The paper does not bring any technical novelty or significant technical advancement (which on itself would not be a reason to reject from TMLR), but it neither brings any strong theoretical analysis (such as subset discovery guaranties, solution identifiability, etc.) - this part of audience would hence not find the paper of any particular value. The high computation cost of the method limits its practical applicability which is documented by the presented experiments being conducted on fairly small scale datasets - this means little use to practitioners. The paper could potentially be of most interest to the XAI part of the community but this is somehow lost in the narrative. As in the comment above, the paper could greatly benefit from re-thinking its main messages, target audience and hence the overall narrative.

**Claims And Evidence:**

No

**Claims Explanation:**

While there is no clear gap between the presented claims and provided evidence, the reviewers agree that the overall narrative of the paper is unclear and needs fundamental revision. While the title may guide the reader to assume the paper discusses a Rashomon algorithm targeting the estimation of the Rashomon sets, the paper does not in reality aspire to do this. Instead it rephrases the rather well-known problem of feature selection instability, especially in overparametrized settings, as the exploration of the Rashomon effect. While noting this connection is well justified, it is not clear what does it actually bring theoretically or practically. The interpretation of the presented SWAG algorithm as a Rashomon algorithm is therefore somewhat misleading. On the other hand, the algorithm is very clearly related to methods for variable selection, many of which acknowledge the possible existence of multiple nearly equivalent models with different sub-sets of selected variables and the consequence of these phenomenon on the interpretability of the models. These links are, however, not clearly covered in the main paper and the discussion of the state of the art in this area is relegated to the appendix making it more difficult for the reader to follow and place the work into the relevant context. Moreover, as the authors correctly acknowledge, the body of works in variable selection is vast and covers many concepts mentioned in the current paper - wrapper methods, stability of variable selection, identifiability of solutions, etc. These are, however, not sufficiently discussed in the paper with respect to the the state of the art in the variable selection are of work. All the above leads to the paper overall significantly lacking in clarity of the narrative and argumentation. I recommend the authors rethink the positioning of the paper and its contribution with respect to the existing research especially in view of similar concepts being discussed substantially in the variable selection domain.

**Resubmission Of Major Revision:**

The authors may consider submitting a major revision at a later time.